# Efficient Time Series Processing for Transformers and State-Space Models through Token Merging

**Leon Götz** [1 2]  **Marcel Kollovieh** [2 3 4]  **Stephan Günnemann** [2 3 4]  **Leo Schwinn** [2 3]

## Abstract

Despite recent advances in subquadratic attention mechanisms or state-space models, processing long token sequences still imposes significant computational requirements. Token merging has emerged as a solution to increase computational efficiency in computer vision architectures. In this work, we perform the first investigations of token merging in *time series analysis* on both transformers and state-space models. We further introduce *local merging*, a domain-specific token merging algorithm that selectively combines tokens within a local neighborhood, achieving two major benefits: a) Local merging can adjust its computational complexity from quadratic to linear based on the neighborhood size to effectively scale to long sequences; b) Local merging is the first causal merging scheme enabling token merging in transformer decoders. Further, we identify spectral properties of the input data that reliably predict the potential benefits of local merging without requiring evaluation on downstream tasks. Our comprehensive empirical evaluation demonstrates that local merging offers substantial efficiency gains with minimal impact on accuracy, achieving up to $5400\%$ acceleration on the recently proposed Chronos foundation model.

## 1. Introduction

Since their inception in NLP (Vaswani et al., 2017), transformers have extended their influence into various domains, including computer vision with Vision Transformers (ViTs) (Dosovitskiy et al., 2021), graphs (Yun et al., 2019), and time series processing (Li et al., 2019). However, the computational complexity of the standard attention mecha-

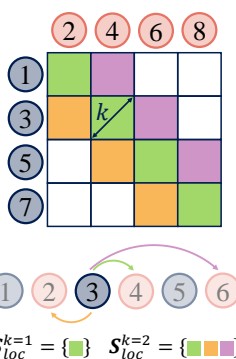

$$S_{loc}^{k=1} = \{ \blacksquare \} \quad S_{loc}^{k=2} = \{ \blacksquare\blacksquare \}$$

Figure 1: **Local token merging:** Computing token similarity on a subset $\mathbf{S}_{loc}$ under locality constraint $k$ reduces token merging's quadratic complexity to linear.

nism scales quadratically with the number of input tokens, resulting in high memory requirements. This scalability issue becomes especially pronounced in time series processing, where sequences frequently comprise thousands of tokens (Godahewa et al., 2021). Consequently, recent foundational models in time series, such as Chronos, exhibit impressive zero-shot generalization capabilities but demand substantial computational resources (Ansari et al., 2024).

Recently, state-space models have emerged as a solution to mitigate the computational burden of transformers. Their complexity scales subquadratically with the sequence length (Poli et al., 2023), which allows them to process millions of tokens (Nguyen et al., 2023). However, even in state-space models, very long sequences will impose considerable memory and computational demands.

Bolya et al. (2023), have shown that the efficiency of ViTs can be substantially improved by *merging* tokens throughout the transformer architecture. Specifically, they compute similarity scores between tokens and combine them into single tokens through a convex combination. However, they only explore token merging for ViT architectures.

In this work, we for the first time explore token merging within the time series domain. We introduce a novel *local* token merging algorithm whose computational complexity varies from quadratic to linear, based on the neighborhood considered for each token merge. This allows token merging to scale to long sequences and be applicable to state-space models. Further, our *local* merging preserves causality and

---

[1]Volkswagen AG [2]Technical University of Munich [3]Munich Data Science Institute [4]Munich Center for Machine Learning. Correspondence to: Leon Götz <leon.goetz@volkswagen.de>.

is the first viable token merging scheme for transformer decoders. The algorithm is illustrated in figure 1. Through comprehensive empirical evaluations, we analyze the impact of token merging on various time series transformer models and state-space models. Our key contributions are:

**Token merging in time series** We extend token merging from computer vision to time series analysis and propose local merging as a domain-specific token merging algorithm.

**Model acceleration** Across five time series transformer architectures (5.1), foundation models (5.3), two state-space models (5.4), and six datasets, token merging reveals substantial computational savings with only slight reductions in accuracy. In some settings, it even improves forecasting performance while accelerating models simultaneously. Token merging enhances model throughput by up to $5400\,\%$ and improves forecasting performance by up to $9\,\%$.

**Token merging outcomes** We identify three distinct outcomes when using token merging: 1) a consistent decline in performance when merging more tokens, 2) initial improvements in accuracy with few merged tokens followed by a drop as merging increases, and 3) scenarios where accuracy remains unchanged regardless of the token merging rate.

**Understand token merging** Our detailed analysis reveals that token merging acts as an adaptive low-pass filter, selectively reducing noise. We further identify model- and dataset-specific properties explaining the effectiveness of our token merging algorithms.

## 2. Related work

**Time series transformers** Recently, many transformer architectures with inductive biases for time series have been proposed. Most of them reduce complexity by modifying the attention mechanism. Informer uses ProbSparse attention (Zhou et al., 2021), while Autoformer leverages autocorrelation as a sequence-based similarity measure (Wu et al., 2021). FEDformer uses the frequency domain to model time series effectively (Zhou et al., 2022). Non-stationary Transformers mitigate the effect of the time series distribution changing over time (Liu et al., 2022b).
Due to their success in the vision and NLP domain, transformer-based foundation models have lately emerged for time series, often used in zero-shot settings. Many works focus on training transformers directly on large and diverse time series datasets, usually with billions of tokens (Garza & Mergenthaler-Canseco, 2023; Das et al., 2023; Rasul et al., 2023; Woo et al., 2024). Inspired by the success of foundation models in NLP, the recently proposed Chronos model converts continuous time series data into a fixed vocabulary (Ansari et al., 2024).

**State-space models** Due to the quadratic scaling of the attention mechanism, transformer architectures suffer from significant computational cost when processing long sequences. Recently, state-space models have shown promising results in overcoming this challenge. Linear state-space layers solve the sequential processing requirement of RNNs (Gu et al., 2021). The S4 model reduces memory requirements by conditioning the state-space matrix with a low-rank correction (Gu et al., 2022). By using implicit convolutions and a data-aware gating mechanism, Hyena (Poli et al., 2023) became one of the first state-space model architectures to match transformers on NLP tasks. Later work uses hardware-aware algorithms to improve the performance on modern accelerators (Gu & Dao, 2023).

**Reducing tokens** Many works reduce the number of processed tokens to increase the efficiency of transformers in computer vision and NLP, often by pruning (Meng et al., 2022; Goyal et al., 2020). Marin et al. (2021) merge tokens in ViT architectures to reduce the loss of information associated with pruning. Bolya et al. (2023) enhance the token merging algorithm, which they successfully apply to already trained encoder-only models. Besides initial work on classification tasks (Bolya et al., 2023), subsequent work applies token merging to diffusion models (Bolya & Hoffman, 2023). Kim et al. (2024) combine merging and pruning, while other works investigate optimal merging and pruning rates (Bonnaerens & Dambre, 2023; Chen et al., 2023). Concurrent work adapts token merging to preserve the spectral properties of the token space (Tran et al., 2024). However, their merging algorithm still has quadratic complexity, making it unsuitable for long sequence processing.

**Sparse attention and token skipping** Besides reducing the number of tokens, sparse attention (Child et al., 2019) and token skipping (Raposo et al., 2024) also decrease the computational requirements of transformer models. In contrast to token merging, sparse attention can only accelerate the attention mechanism itself and not the subsequent MLP, which can take over $60\,\%$ of the total computation (Marin et al., 2021). Concurrent work, such as token skipping (Raposo et al., 2024), involves the selection of a subset of tokens to be processed in a transformer layer. However, it has only been shown in NLP when training from scratch. Token merging, however, can accelerate already trained models and does not require any training data or fine-tuning. This is especially important for recent foundation models. In our experiments in sections 5.1 and 5.2, token merging successfully accelerates Informer and Autoformer, which already employ sparse attention. We therefore consider token merging as an orthogonal approach.

Here, we propose the first token merging algorithm for the time series domain, which extends beyond previous investigations in ViTs (Bolya et al., 2023; Bolya & Hoffman,

2023). We systematically evaluate the potential to reduce computational effort in time-series-specific architectures. See appendix A for more details on related work.

## 3. Token merging

Despite recent advances in efficient transformers, processing long input sequences still induces considerable memory requirements and computational effort. To address this, we first extend token merging, which successfully boosts throughput in computer vision, to time series models. Next, we propose local merging, a domain-specific and efficient token merging algorithm for state-space models and long sequence processing. Finally, we introduce causal merging as a special case of local merging to allow for token merging in decoder architectures and propose dynamic merging to further improve token merging in real-world settings.

**Global token merging for time series**   Let a neural network $\mathbf{f}(\mathbf{x}) = \mathbf{\Phi}_L \circ \mathbf{\Phi}_{L-1} \circ \cdots \circ \mathbf{\Phi}_1(\mathbf{x})$ consist of $L$ layers denoted as $\mathbf{\Phi}_l$, where each layer takes the output of the previous layer as input. We assume that the input $\mathbf{x}_l \in \mathbb{R}^{t_l \times d}$ consists of $t_l$ tokens with dimension $d$. Thereby, the input tokens are generated by a tokenizer $\mathbf{g} : \mathbb{R}^z \to \mathbb{R}^{t \times d}$ out of $z$-dimensional input data $\mathbf{u}$. We assume the input $\mathbf{u} \in \mathbb{R}^{m \times n}$ consists of $m$ time stamps with $n$ variates, and $m \cdot n = z$. To improve the computational efficiency of token-based time series models, we extend global token merging from computer vision to the time series domain. Following Bolya et al. (2023), we combine the $r$ most similar tokens in each layer, reducing the tokens to be processed in layer $l+1$ to $t_{l+1} = t_l - r$. For this, we split the set of all tokens into two disjoint subsets $\mathcal{A}, \mathcal{B}$ in alternation to avoid merging conflicts and allow for a parallelized computation of merging correspondences. Here $\mathcal{A}$ and $\mathcal{B}$ contain $t_l/2$ elements each, denoted as $\mathbf{a}_i$ and $\mathbf{b}_j$ respectively. We compute the cosine similarity between **all** tokens in both subsets $\mathbf{S} = (s_{ij})$ and merge the $\mathrm{top}\ r$ most similar correspondences by averaging the tokens accordingly. This results in a **global** token merging algorithm with **quadratic complexity**. Lastly, Bolya et al. (2023) use a fixed $r$ to enable batch processing without needing to pad individual batch elements to the same shape after token reduction. Later, we introduce more adaptive merging schemes through dynamic merging.

**Local token merging for time series**   In this work, we design new token merging mechanisms for time series architectures and demonstrate run-time and even performance improvements over various datasets and models.

Previous work on token merging in image processing explored **global** merging schemes, where every token of each subset $\mathcal{A}$ and $\mathcal{B}$ could be merged with each other (Bolya et al., 2023; Bolya & Hoffman, 2023). However, computing the similarity $\mathbf{S} \in \mathbb{R}^{t_l/2 \times t_l/2}$ between both sets of tokens

has a complexity of $O(t_l^2/4)$, which is suboptimal for sequential data often consisting of long token sequences (Godahewa et al., 2021; Grešová et al., 2023), and state-space models featuring subquadratic complexity (Poli et al., 2023; Nguyen et al., 2023).

Therefore, we propose **local merging** - a superset of token merging - by introducing $k \in \mathbb{N}, 1 \leqslant k \leqslant t_l/2$ as a locality constraint where we compute the similarity only on a local subset of tokens:

$$\mathbf{S}_{loc} = \{s_{ij} \,|\, 1 \leqslant i, j \leqslant t_l/2, \, |i - j| < k\}. \qquad (1)$$

Figure 1 illustrates the proposed merging algorithm. The locality constraint reduces the complexity to:

$$O(t_l/2 + (k-1)(t_l - k)). \qquad (2)$$

Varying the locality, we achieve **linear complexity** by considering only neighboring tokens for merging up to quadratic complexity by considering a global merging pool, possibly exploiting more redundancy. For efficient computation, we refactor $\mathbf{S}_{loc}$ into a rectangular tensor. An upper bound for the resulting speed-up can be given by speed up $\leqslant 3\,L\,4^{L-1} \cdot (4^L - 1)^{-1}$. The acceleration of deeper models is expected to increase as more subsequent layers can profit from already merged tokens. Local merging additionally preserves order and locality as an inductive bias for sequence processing.

Some time series transformers use processing mechanisms that require a minimum number of tokens in the forward pass. To universally enable token merging in these architectures, we further introduce $q$ as the minimum number of remaining tokens. When encountering odd numbers of tokens $t_l$, we exclude the most recent token from merging, as we expect it to contain the most relevant information following the Markov assumption. We derive the complexity of the token merging procedures in appendix B.1 and further discuss the interplay of time-series-specific inductive biases and token merging in appendix B.2.

**Causal token merging for decoders**   Existing merging schemes are not suitable for causal operations, as global token merging transfers information over arbitrary ranges. To remedy this limitation and enable token merging in transformer decoders, such as for recent decoder-only foundation models (Das et al., 2023) and encoder-decoder architectures (Ansari et al., 2024), we propose a special case of local merging: By restricting the merging neighborhood to only adjacent tokens with $k = 1$, local merging preserves temporal **causality**.

Token merging reduces the number of tokens to be processed throughout the model. However, many architectures require a fixed number of decoder output tokens or fixed dimensions for linear projection output layers. To maintain a constant output dimensionality while merging tokens to speed-up the decoder, we unmerge all tokens in a final

step. Coherent with our causal merging operation, we clone a previously merged token into two neighboring identical ones, to unmerge it. Bolya & Hoffman (2023) propose an unmerging algorithm for computer vision. However, they only leverage non-causal global token merging. Moreover, they immediately unmerge after every merge, making it unsuitable for long sequence processing, as it is unable to utilize the cumulative effect of reducing tokens.

**Dynamic token merging** A fixed merging scheme allows for batch processing without needing to pad individual time series to the same length. However, it enforces a constant $r$ among layers and batches, which might not always be optimal, as dissimilar tokens might be merged. We introduce dynamic merging to further improve token merging in real-world settings, where batch sizes are often small. To this end, we determine the number of tokens to be merged dynamically for every batch element using a token cosine similarity threshold. To avoid padding, we average them throughout the batch. As a result, dynamic merging enables optimal merging rates that are adaptive to the input data and current network layer.

## 4. Experiments

We systematically explore local merging in diverse settings on 5 time series datasets and 5 model architectures in 5 different sizes each. Additionally, we investigate local merging in large foundation models using Chronos in a zero-shot setting (Ansari et al., 2024). Finally, we demonstrate that local merging can be applied to state-space models for long sequence processing. See appendix C for more details on our experimental settings.

**Datasets** We use time series forecasting datasets including ETTh1, ETTm1, Weather, Electricity and Traffic for our transformer experiments. For state-space models, we use the long-range Dummy Mouse Enhancers Ensembl dataset.

**Model architectures** For our main experiments, we use 5 architectures, including Autoformer, FEDformer, Informer, Non-stationary Transformer, and the vanilla Transformer (Vaswani et al., 2017) as reference. For each model, we evaluate local merging for different model sizes with $L \in \{2, 4, 6, 8, 10\}$ encoder layers, which we train doing hyperparameter optimization. We use an input length of $m = 192$, following the results of Nie et al. (2023), and a prediction horizon $p = 96$ samples. Longer sequences would generally benefit token merging.

For experiments on the foundation model Chronos, we use the default input length of $m = 512$ and prediction horizon $p = 64$ (Ansari et al., 2024). We compute the median from Chronos probabilistic forecasts and report the MSE.

For our experiments on state-space models, we use HyenaDNA medium, a genomic foundation model (Nguyen

et al., 2023) based on the Hyena architecture (Poli et al., 2023) and Mamba (Gu & Dao, 2023) models with the same hyperparameters as Hyena. We use a large input length of $m = 16\,000$ nucleotides, utilizing state-space models' subquadratic complexity.

**Applying local merging** Allowing self-attention to transfer information between tokens before merging them is beneficial in our experiments. Therefore, we apply local merging after the self-attention in transformer architectures as Bolya et al. (2023). In state-space models, we merge tokens after the Hyena or Mamba operator.

**Reproducibility of measurements** We report all results on the same Nvidia A6000 GPU and do multiple measurements to achieve inference time standard deviations $< 2\,\%$.

## 5. Results

We first present our main results for local merging on pretrained models and models trained with local merging. Next, we scale local merging to large foundation models and explore token merging for state-space models and long sequence processing. Finally, we investigate if we can gain even higher speed-ups in real-world settings, leveraging dynamic merging schemes.

### 5.1. Local merging in pretrained models

We investigate local merging in both the encoder and decoder on diverse time series transformer models with different inductive biases. All models are trained on the target dataset and local merging is applied only during inference time, as accelerating already trained models is of high practical relevance. We choose local merging hyperparameters as described in appendix C, selecting the fastest token merging trial on the validation set that is within a $0.01$ increase in MSE compared to the reference without token merging. If we do not find a trial with token merging satisfying these tight criteria, we report results without token merging, mimicking how local merging might be applied in practice. We perform all selections on the validation set and report all results on the test set.

The vanilla and Non-stationary Transformers have quadratic attention mechanisms, while the remaining architectures feature subquadratic attention complexities of $O(t_l \cdot \log(t_l))$ for Autoformer and Informer and $O(t_l)$ for FEDformer. Regardless, our local merging in the encoder together with our casual merging in the decoder substantially increase the throughput of most models, up to $3.80\times$, often with no change in forecasting quality, as table 1 shows. In some experiments, local merging even improves the MSE. In line with our formal analysis of potential speed-up from token merging in section 3, we generally observe higher accelerations for larger models, as more subsequent layers can

profit from already merged tokens. Independent of model size, local merging finds Pareto optimal points in 17 of 25 settings and has no negative effect in the remaining cases. In some cases, we do not find a model with decent forecasting quality satisfying our criteria. Here, token merging during test only has a larger impact on model accuracy, such as for Autoformer on the Traffic dataset. We address this issue when training with token merging in section 5.2.

### 5.2. Local merging during training

Here, we apply local merging during training to reduce the models' sensitivities to the algorithm at inference time. As shown in figure 2, models trained with token merging often outperform those trained without it, even if token merging is not applied during testing. This approach enables us to accelerate models such as Autoformer on the Traffic dataset without sacrificing accuracy, which was previously not feasible when applying token merging only during inference. Additionally, local merging accelerates the training process itself by up to $2.27\times$ for Autoformer on the Traffic dataset.

### 5.3. Scaling to large models

Foundation models are getting more relevant across domains, including NLP (Touvron et al., 2023), computer vision (Kirillov et al., 2023), and time series processing (Das et al., 2023). However, these models have high compu-

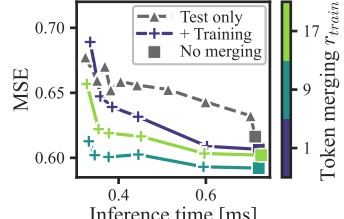 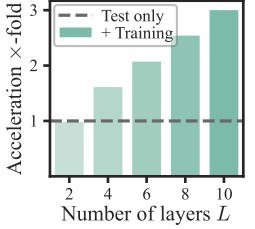

(a) Non-stationary 6 layers on Traffic   (b) Autoformer on Traffic

Figure 2: **(a)** Training with different token merging $r_{train}$ fractions compared to applying token merging only during inference. **(b)** Models that showed high MSE degradation with token merging without training show high accelerations while maintaining MSE when enabling token merging during training.

tational requirements. Therefore, accelerating foundation models without the need for additional fine-tuning is especially important. Thus, we investigate local merging for foundation models on Chronos, a univariate probabilistic model, in a zero-shot forecasting setting (Ansari et al., 2024) and apply local merging only during inference.

In all our experiments, we find Pareto optimal points with token merging. For four out of five datasets, local merging improves both accuracy and throughput simultaneously (see appendix D.1). Our results demonstrate that it is often beneficial to choose a larger Chronos model with token merging over a smaller one without, as in figure 3. We report our results in table 2, choosing the best model without token

Table 1: Local merging accelerates various pretrained transformers of different sizes on several multivariate time series datasets. Merging induces minimal change in quality ($\text{MSE}_\Delta$) compared to the reference without token merging (MSE).

| Dataset | Layers $L$ | Transformer | | | Autoformer | | | FEDformer | | | Informer | | | Non-stationary | | |
|---|---|---|---|---|---|---|---|---|---|---|---|---|---|---|---|---|
| | | MSE | Accel. | $\text{MSE}_\Delta$ | MSE | Accel. | $\text{MSE}_\Delta$ | MSE | Accel. | $\text{MSE}_\Delta$ | MSE | Accel. | $\text{MSE}_\Delta$ | MSE | Accel. | $\text{MSE}_\Delta$ |
| ETTh1 | 2 | 0.75 | 1.38× | 0% | 0.42 | 1.00× | 0% | 0.38 | 1.29× | 0% | 0.87 | 1.40× | 0% | 0.55 | 1.36× | 0% |
| | 4 | 0.71 | 1.81× | 0% | 0.40 | 1.39× | 1% | 0.39 | 1.74× | 0% | 0.92 | 1.30× | 1% | 0.47 | 1.82× | 2% |
| | 6 | 0.66 | 2.33× | 0% | 0.44 | 2.12× | 0% | 0.38 | 2.27× | 0% | 0.93 | 2.39× | 0% | 0.46 | 2.39× | 0% |
| | 8 | 0.84 | 2.90× | 0% | 0.41 | 2.68× | −5% | 0.39 | 2.81× | 0% | 1.23 | 2.20× | 9% | 0.48 | 2.93× | 0% |
| | 10 | 0.69 | 3.51× | 0% | 0.39 | 3.14× | 0% | 0.38 | 3.36× | 0% | 1.16 | 2.45× | 4% | 0.57 | 3.56× | 0% |
| ETTm1 | 2 | 0.52 | 1.35× | 0% | 0.44 | 1.00× | 0% | 0.36 | 1.00× | 0% | 0.65 | 1.40× | 0% | 0.42 | 1.36× | 0% |
| | 4 | 0.58 | 1.85× | 2% | 0.43 | 1.00× | 0% | 0.37 | 1.76× | 2% | 0.60 | 1.78× | −1% | 0.48 | 1.72× | 0% |
| | 6 | 0.62 | 2.11× | 4% | 0.45 | 1.00× | 0% | 0.38 | 1.00× | 0% | 0.59 | 2.16× | −1% | 0.38 | 2.52× | 0% |
| | 8 | 0.60 | 3.09× | 1% | 0.58 | 2.60× | 0% | 0.33 | 1.00× | 0% | 0.61 | 1.61× | 0% | 0.46 | 2.10× | −2% |
| | 10 | 0.62 | 3.72× | 0% | 0.54 | 1.69× | 0% | 0.36 | 1.00× | 0% | 0.57 | 1.00× | 0% | 0.41 | 3.80× | 0% |
| Weather | 2 | 0.25 | 1.44× | −1% | 0.28 | 1.10× | 0% | 0.27 | 1.37× | −2% | 0.35 | 1.43× | −1% | 0.19 | 1.46× | 1% |
| | 4 | 0.28 | 1.95× | 0% | 0.24 | 1.00× | 0% | 0.26 | 1.74× | 0% | 0.24 | 1.89× | 2% | 0.19 | 1.95× | 0% |
| | 6 | 0.28 | 2.19× | 9% | 0.26 | 2.03× | 2% | 0.27 | 2.42× | 0% | 0.21 | 2.19× | 2% | 0.20 | 2.54× | 0% |
| | 8 | 0.32 | 2.20× | 5% | 0.26 | 1.56× | 4% | 0.27 | 2.88× | 0% | 0.30 | 1.56× | 1% | 0.20 | 3.14× | 0% |
| | 10 | 0.35 | 2.49× | 8% | 0.26 | 1.72× | 3% | 0.24 | 1.00× | 0% | 0.31 | 1.69× | 1% | 0.19 | 3.76× | 0% |
| Electricity | 2 | 0.25 | 1.30× | 0% | 0.18 | 1.00× | 0% | 0.20 | 1.24× | 0% | 0.30 | 1.23× | 8% | 0.17 | 1.31× | 0% |
| | 4 | 0.26 | 1.75× | 0% | 0.19 | 1.00× | 0% | 0.19 | 1.64× | 0% | 0.30 | 1.60× | 7% | 0.17 | 1.73× | 1% |
| | 6 | 0.25 | 2.29× | 0% | 0.19 | 1.00× | 0% | 0.20 | 2.22× | 0% | 0.29 | 1.00× | 0% | 0.17 | 2.26× | 0% |
| | 8 | 0.25 | 2.84× | 0% | 0.19 | 1.00× | 0% | 0.20 | 2.72× | 0% | 0.31 | 1.00× | 0% | 0.17 | 2.76× | 0% |
| | 10 | 0.25 | 3.31× | 0% | 0.18 | 1.00× | 0% | 0.20 | 3.33× | 0% | 0.30 | 1.00× | 0% | 0.18 | 2.53× | 7% |
| Traffic | 2 | 0.66 | 1.28× | 1% | 0.63 | 1.00× | 0% | 0.59 | 1.21× | 0% | 0.68 | 1.19× | 6% | 0.60 | 1.27× | 2% |
| | 4 | 0.66 | 1.56× | 3% | 0.60 | 1.00× | 0% | 0.58 | 1.65× | 0% | 0.68 | 1.00× | 0% | 0.59 | 1.68× | 1% |
| | 6 | 0.64 | 2.13× | 1% | 0.61 | 1.00× | 0% | 0.57 | 2.10× | 0% | 0.69 | 1.00× | 0% | 0.62 | 1.58× | 2% |
| | 8 | 0.68 | 2.67× | 0% | 0.60 | 1.00× | 0% | 0.59 | 2.61× | 0% | 0.71 | 1.00× | 0% | 0.59 | 2.69× | 1% |
| | 10 | 0.67 | 3.25× | −1% | 0.59 | 1.00× | 0% | 0.58 | 3.12× | 0% | 0.69 | 1.00× | 0% | 0.59 | 3.16× | 0% |

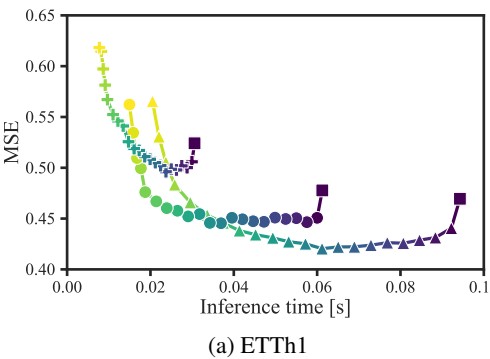
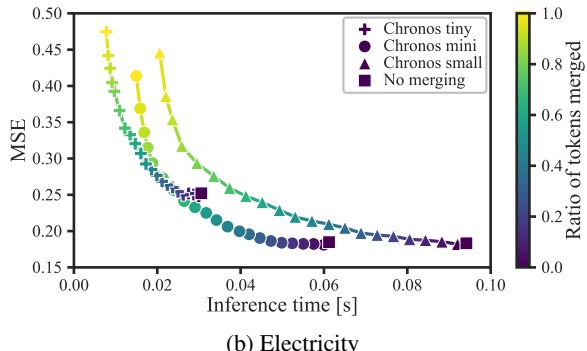

(a) ETTh1              (b) Electricity

Figure 3: MSE for different token merging in Chronos models during zero-shot testing on two datasets. Choosing larger models with token merging is beneficial compared to smaller ones without.

merging as reference. We illustrate two cases: 1) Selecting the token merging setting that provides the best MSE, and 2) selecting the setting with the fastest throughput. For 2), we constrain the MSE of token merging trials to be lower than the second-best model without token merging. In addition, we allow a maximum increase in MSE of 3 % compared to the reference. In our experiments, we can improve Chronos MSE by up to 9 % and speed-up inference by $54.76\times$.

Table 2: Local merging accelerates all Chronos foundation models from tiny to large during zero-shot forecasting. Applying local merging, we aim for two objectives: best MSE and fastest acceleration. Among Chronos models, we choose the best without token merging as reference (MSE). As local merging improves MSE (negative $\text{MSE}_\Delta$) while speeding up models, we are able to choose small models while surpassing forecasting quality of larger ones.

| Dataset | MSE | Best | | Fastest | |
|---|---|---|---|---|---|
| | | Accel. | $\text{MSE}_\Delta$ | Accel. | $\text{MSE}_\Delta$ |
| ETTh1 | 0.45 | $14.17\times$ | $-6\,\%$ | $32.76\times$ | $2\,\%$ |
| ETTm1 | 0.41 | $1.23\times$ | $-4\,\%$ | $6.47\times$ | $3\,\%$ |
| Weather | 0.17 | $1.16\times$ | $-1\,\%$ | $54.76\times$ | $3\,\%$ |
| Electricity | 0.14 | $1.02\times$ | $0\,\%$ | $2.91\times$ | $3\,\%$ |
| Traffic | 0.61 | $1.16\times$ | $-9\,\%$ | $2.91\times$ | $1\,\%$ |

## 5.4. Token merging in state-space models

State-space models can process very long sequences with millions of tokens due to their subquadratic complexity. Our proposed local merging algorithm is specifically designed to match this subquadratic complexity, enabling effective token merging in state-space models. Additionally, it preserves locality and order as inductive bias for sequence processing. We compare local and global token merging in HyenaDNA (Grešová et al., 2023) and Mamba (Gu & Dao, 2023), for two objectives: the largest speed-up and the best prediction quality. We use a classification task, where the data consists of long genomic sequences with $16\,000$ nucleotides each. Our local merging with $k = 1$ featuring linear complexity and locality bias outperforms global merging with $k = t_l/2$ and quadratic complexity. Table 3

illustrates that local merging achieves substantially larger speed-up and better accuracy than global merging. This experiment indicates that architecture and domain-specific biases are important when applying token merging. Local merging accelerates HyenaDNA up to $3.62\times$ with a $4.9\,\%$ decrease in accuracy, whereas global merging substantially reduces the accuracy by $9.5\,\%$. Utilizing less aggressive merging schemes, local merging even boosts accuracy by $1.7\,\%$ while still accelerating HyenaDNA $1.68\times$. In Mamba models, local merging achieves even higher accelerations up to $4.09\times$. In our experiments, the similarity computation of local merging adds $14\,\%$ of additional execution time to every Hyena block. For global merging, however, this is substantially higher with $68\,\%$, highlighting the importance of local merging's linear complexity. To our knowledge, this is the first study of merging individual states in state-space models to improve their sequence modeling performance.

Table 3: Comparison of **global** and **local** token merging for Hyena and Mamba on the long sequence Dummy Mouse Enhancers Ensembl dataset. **Best**, second.

| Token merging | Hyena | | Mamba | |
|---|---|---|---|---|
| | Accel. | Accuracy | Accel. | Accuracy |
| No merging | $1.00\times$ | $78.9\,\%$ | $1.00\times$ | $76.0\,\%$ |
| Local merging[fastest] | $\underline{3.62\times}$ | $74.0\,\%$ | $\mathbf{4.09\times}$ | $74.4\,\%$ |
| Local merging[best] | $1.68\times$ | $\mathbf{80.6\,\%}$ | $1.65\times$ | $76.0\,\%$ |
| Global merging[fastest] | $2.93\times$ | $69.4\,\%$ | $2.81\times$ | $74.0\,\%$ |
| Global merging[best] | $1.15\times$ | $\underline{80.2\,\%}$ | $1.27\times$ | $76.4\,\%$ |

## 5.5. Dynamic token merging

Dynamic token merging mitigates the issue of dissimilar tokens being merged, potentially improving quality. Here, we leverage the single-sample case with batch size 1 to explore dynamic merging in optimal conditions. From a practical perspective, this case might be relevant for on-device applications like smartphones or automated driving. We further utilize small batch sizes of 10 elements as they might appear in real-world applications.

In figure 4, we compare token merging utilizing a fixed $r$ to dynamic merging, varying the cosine similarity threshold. Dynamic merging improves quality slightly in most settings. With batch size 10, it is marginally worse compared to the optimal single-sample case. Therefore, we suggest using a fixed merging schedule for applications with large batches and dynamic merging for cases with few batch elements. There is no equivalent $r$ to dynamic merging schedules as they are similarity-based and strongly layer-dependent. We report FLOPs as we observe substantial execution overhead in time measurements.

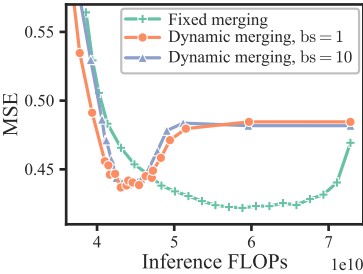

Figure 4: Comparison of dynamic merging based on a similarity threshold with fixed $r$ merging for Chronos small on ETTh1.

# 6. Further investigations

We first explore different local merging outcomes regarding acceleration and forecasting performance. Next, we find dataset- and model-specific properties explaining why, in some cases, local merging can improve forecasting quality. Lastly, we investigate if we can achieve same effects on model acceleration and forecasting performance with simpler methods than local merging. We further explore different token similarity measures in appendix E.1, compare merging with pruning in appendix E.2, and investigate the influence of tokenization on local merging in appendix E.3.

## 6.1. Merging outcomes

We observe three distinct merging outcomes when combining tokens in transformer architectures.

**Increasing MSE** As the number of merged tokens increases, the MSE increases almost monotonically (see figure 3b). This behavior can be explained due to a loss of information when combining multiple tokens and also occurs in the vision domain (Bolya et al., 2023).

**Constant MSE** For the vanilla Transformer on ETTh1 and for FEDformer on ETTh1, Weather, Electricity, and Traffic, we observe a constant MSE when applying token merging as shown in figure 5. For the Transformer model, we find all tokens to be similar after the first attention block. Thus, token merging does not affect the model performance.

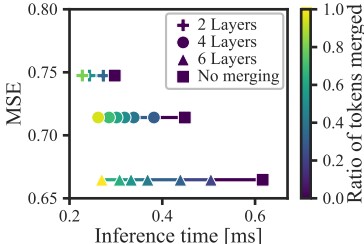

Figure 5: Transformer models on ETTh1 show constant MSE, independent of the amount of token merging $r$.

Nevertheless, we find that in most cases, these models still provide reasonable forecasts. In our experiments, transformer models trained on larger or more complex datasets containing more variates do not show this behavior. We argue that this might be a limitation of transformers on small time series datasets (Zeng et al., 2023; Li et al., 2023). Still, token merging successfully improves the throughput while maintaining accuracy for these models.

**Decreasing MSE** Token merging increases forecasting quality, most prominently in Chronos models, as in figure 3a. We explain this behavior in section 6.2.

## 6.2. Improvement of forecasting quality

We explore if the potential benefits of token merging can be predicted through dataset- and model-specific properties without requiring downstream task evaluation.

**Selective low-pass filter** We hypothesize that local merging improves forecasting quality by selectively reducing noise. Averaging similar tokens smoothes the time series, acting as an adaptive low-pass filter. To validate our hypothesis, we low-pass filter the input time series using Gaussian kernels without token merging in figure 6. On ETTh1, both local merging and Gaussian filtering improve the MSE. On the Electricity dataset, token merging and Gaussian filtering do not positively impact the MSE. All of these observations are in line with our hypothesis. Applying token merging together with the Gaussian kernel leads to the best results. Other averaging kernels were significantly worse. We show additional results in appendix E.4.

**Dataset properties** We find properties of the target dataset that are particularly amenable to token merging. Using metrics from signal processing, we can predict how well local merging will perform on a new dataset prior to evaluation. Improvement in forecasting quality due to local merging in table 2 correlates with the spectral entropy of the dataset. Specifically, local merging achieves higher quality gains on high entropy datasets, such as ETTm1, ETTh1, and Traffic (see table 4). We argue that local merging removes unnecessary information from complex signals with high entropy using its selective smoothing ability. This allows the model to focus on only the relevant patterns of a signal and to

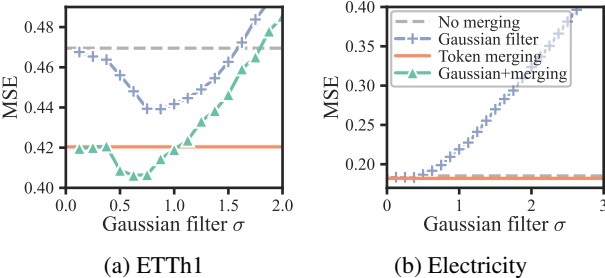

(a) ETTh1  (b) Electricity

Figure 6: Comparison of low-pass filtering the input time series with a Gaussian filter and token merging for Chronos small. The Gaussian filter has a similar effect on MSE as local merging, supporting our hypothesis that local merging selectively low-pass filters data. Besides improving MSE, local merging accelerates models unlike Gaussian filtering.

achieve better prediction quality. Besides the spectral entropy, the same correlation is evident in the total harmonic distortion (THD). Local merging adaptively low-pass-filters noisy distorted signals to condense the most relevant patterns and effectively improves the signal-to-noise ratio. The greater noise in ETTm1, ETTh1, and Traffic compared to Electricity and Weather can also be visually inspected in the respective frequency spectrum in appendix E.5. Therefore, we expect a larger improvement of prediction quality when applying local merging on high entropy signals with a low signal-to-noise ratio. While less significant, local merging still improves efficiency on low noise datasets, as we discuss in appendix E.6 in detail.

Table 4: Quality improvement due to local merging on datasets with different signal properties.

| Dataset | MSE$_\Delta$ | Spectral entropy | THD |
|---|---|---|---|
| ETTm1 | $-4\%$ | 4.64 | 70.23 |
| ETTh1 | $-6\%$ | 4.55 | 54.93 |
| Traffic | $-9\%$ | 2.96 | 19.78 |
| Electricity | $0\%$ | 2.24 | 15.77 |
| Weather | $-1\%$ | 1.64 | 13.15 |

**Model properties**  Across all datasets, we identify model-specific properties that benefit local merging. For this, we analyze the average cosine similarity of tokens in the models from table 1 after the first transformer layer. Local merging accelerates models, such as the Non-stationary Transformer, that learn more similar token representations without quality degradation. For models with dissimilar token representations, like the Informer, we observe quality degradations when applying local merging, as table 5 shows.

### 6.3. Dependencies on input length

Token merging effectively reduces the number of tokens in a transformer layer. Here, we explore if we can achieve similar accelerations while maintaining the same prediction quality by varying the number of input samples $m$. For

Table 5: Quality degradation due to local merging of models with different token representations.

| Model and dataset | MSE$_\Delta$ | Token similarity |
|---|---|---|
| Informer 2 Layers Traffic | $6\%$ | 0.10 |
| Informer 4 Layers Electricity | $7\%$ | 0.22 |
| Informer 8 Layers ETTh1 | $9\%$ | 0.28 |
| Informer 6 Layers Weather | $2\%$ | 0.35 |
| Informer 6 Layers ETTm1 | $-1\%$ | 0.40 |
| Non-stationary 10 Layers ETTh1 | $0\%$ | 0.77 |
| Non-stationary 8 Layers ETTh1 | $0\%$ | 0.82 |
| Non-stationary 6 Layers Weather | $0\%$ | 0.87 |
| Transformer 10 Layers ETTm1 | $0\%$ | 0.99 |

better comparison, we keep the predicted time series snippet fixed and only adjust the input sequence.

Our results demonstrate that varying the input length cannot replace local merging (see also appendix E.7). In figure 7, we investigate input length dependence for two objectives in more detail: First, we explore the token merging setup that leads to the best MSE and compare the results to the model without merging. Here, local merging yields considerable throughput increases while improving predictive quality at the same time. Second, we compare the fastest model with token merging, which shows no quality decreases, to a standard model. We find models with local merging to scale favorable to long sequences.

We further explore the redundancy of input tokens including the influence of the positional embedding in appendix E.6.

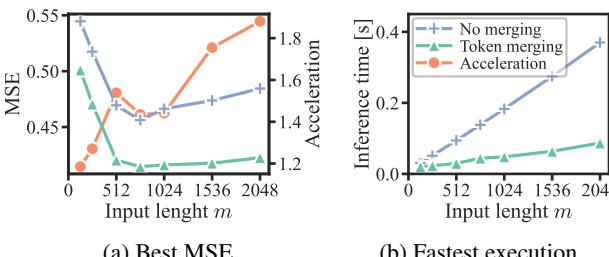

(a) Best MSE  (b) Fastest execution

Figure 7: Effect of input length on **(a)** forecasting quality and **(b)** inference time for token merging in Chronos small on ETTh1.

## 7. Conclusion

In this work, we explore token merging in the time series domain for the first time. We conduct an extensive empirical study on transformer architectures and state-space models in diverse settings using various models and datasets. We demonstrate that token merging can successfully accelerate pretrained models and sometimes even improve their prediction quality. We further introduce a domain-specific *local merging* algorithm with variable complexity and illustrate its effectiveness on Hyena and Mamba models. Additionally, local merging is the first causal token merging scheme,

which we successfully apply in transformer decoders. Finally, we conduct several ablation studies to investigate when token merging is most effective, including spectral properties of the analyzed dataset and model- and algorithm-specific properties. We hope that token merging will have a positive effect on reducing the resource consumption and environmental impact of time series models.

**Limitations** In our work, we divide all tokens into two sets and restrict merging to occur only between tokens from different sets. Future work can explore more flexible merging schemes for time-series-specific architectures. Moreover, we do not conduct ablations on all possible hyperparameters due to the large number of architectures and datasets evaluated in this work.

## Impact Statement

We demonstrate large accelerations and considerable quality gains throughout a broad range of time series architectures. Our local merging can improve training efficiency and accelerate already trained models without any fine-tuning. This is especially important for emerging foundation models, which require considerable computational resources. We hope that local merging will contribute to more sustainable time series models and reduce their environmental impact.

**Disclaimer** The results, opinions, and conclusions expressed in this publication are not necessarily those of Volkswagen Aktiengesellschaft.

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

## A. Related work

Here, we discuss related work in greater detail.

**Time series transformers**   In recent years, many transformer architectures with inductive biases for time series have been proposed, successfully outperforming classical and other deep-learning-based methods in time series forecasting quality like recurrent neural networks (Li et al., 2019). Most of them focus on reducing complexity by modifying the attention mechanism. LogTrans uses LogSparse attention (Li et al., 2019), while Informer focuses only on the most relevant queries using ProbSparse attention (Zhou et al., 2021). Additionally, many architectures adopt decomposition techniques to model trend and seasonal patterns (Woo et al., 2022; Wu et al., 2021; Zhou et al., 2022; Liu et al., 2022b). Autoformer leverages autocorrelation as a sequence-based similarity measure in the attention mechanism (Wu et al., 2021). FEDformer uses the frequency domain to model time series effectively (Zhou et al., 2022). Non-stationary Transformers further mitigate the effect of the time series distribution changing over time (Liu et al., 2022b). Other works apply hierarchical attention (Liu et al., 2022a; Cirstea et al., 2022), embed subsequences as tokens to capture local semantic information (Nie et al., 2023), or leverage attention between time series variates to better model multivariate patterns (Zhang & Yan, 2023; Liu et al., 2023). Due to their success in the vision and NLP domain, transformer-based foundation models have lately emerged for time series, often used in zero-shot settings. Many works focus on training transformers directly on large and diverse time series datasets, usually with billions of tokens (Garza & Mergenthaler-Canseco, 2023; Das et al., 2023; Rasul et al., 2023; Woo et al., 2024). Inspired by the success of foundation models in NLP, the recently proposed Chronos model converts continuous time series data into a fixed vocabulary and is trained on both real-world and synthetic data (Ansari et al., 2024). Besides, other research branches focus on fine-tuning vision or NLP models for time series (Zhou et al., 2023) and on applying large language models directly on time series data (Gruver et al., 2023).

**State-space models**   Due to the quadratic scaling of the attention mechanism, transformer architectures suffer from significant computational cost when processing very long sequences. Recently, state-space models have shown promising results in overcoming the quadratic complexity of transformers with respect to input length. Linear state-space layers solve the sequential processing requirement of RNNs through linear state-space representations (Gu et al., 2021). The S4 model reduces memory requirements by conditioning the state-space matrix with a low-rank correction (Gu et al., 2022). By using implicit convolutions and a data-aware gating mechanism, Hyena (Poli et al., 2023) became one of the first state-space model architectures to match transformers on NLP tasks. Later work uses hardware-aware algorithms to improve the performance of state-space models on modern accelerators (Gu & Dao, 2023).

**Reducing tokens**   Many works reduce the number of processed tokens to increase the efficiency of transformers in computer vision and NLP, often by pruning (Meng et al., 2022; Goyal et al., 2020). Marin et al. (2021) merge tokens in ViT architectures to reduce the loss of information associated with pruning. Bolya et al. (2023) enhance the token merging algorithm, which they successfully apply to already trained encoder-only models. Besides initial work on classification tasks (Bolya et al., 2023), subsequent work applies token merging to diffusion models (Bolya & Hoffman, 2023). Kim et al. (2024) combine merging and pruning, while other works investigate optimal merging and pruning rates (Bonnaerens & Dambre, 2023; Chen et al., 2023). Concurrent work adapts token merging to preserve the spectral properties of the token space (Tran et al., 2024). However, their merging algorithm still has quadratic complexity, making it unsuitable for long sequence processing.

**Sparse attention and token skipping**   Besides reducing the number of tokens, sparse attention (Child et al., 2019; Li et al., 2019; Zhou et al., 2021; Wu et al., 2021) and token skipping (Raposo et al., 2024) also decrease the computational requirements of transformer models. Sparse attention computes a subset of the attention matrix. Therefore, it can only accelerate the attention mechanism itself and not the subsequent MLP, in contrast to reducing the number of tokens during token merging. According to Marin et al. (2021), this MLP can take over $60\,\%$ of the total computation in a ViT layer. Further, altering the network architecture from full attention to sparse attention might require a retraining of the model. Concurrent work, such as token skipping (Raposo et al., 2024), involves the selection of a subset of tokens to be processed in a transformer layer. However, it has only been shown in NLP when training from scratch. In contrast to sparse attention and token skipping, token merging can accelerate already trained models and does not require any training data or fine-tuning. This is especially important for recent foundation models, which are expensive to train. In our experiments in sections 5.1 and 5.2, token merging successfully accelerates Informer and Autoformer, which already employ sparse attention. We therefore consider token merging as an orthogonal approach.

Here, we propose the first token merging algorithm for the time series domain, which extends beyond previous investigations of token merging in ViTs (Bolya et al., 2023; Bolya & Hoffman, 2023). We systematically evaluate the potential to reduce computational effort in time-series-specific transformer architectures and state-space models.

## B. Local token merging for time series

In the following, we provide derivations and more details on our local merging algorithm. We further discuss the interplay of time-series-specific inductive biases and our token merging method.

### B.1. Derivations

Here, we derive our theoretical results in section 3.

**Complexity of local merging**    To compute $\mathbf{S}_{loc}$ for local merging we need to compute the main diagonal of $\mathbf{S} \in \mathbb{R}^{t_l/2 \times t_l/2}$ and depending on $k$ also secondary diagonals which are symmetrical but shorter than the main diagonal for $k > 1$. We derive the complexity of local merging depending on $k$ in the following:

$$
\begin{aligned}
\text{complexity } \mathbf{S}_{loc} &= \frac{t_l}{2} + 2 \sum_{p=2}^{k} \frac{t_l}{2} - (p-1) \\
&= \frac{t_l}{2} + 2 \sum_{p=1}^{k-1} \frac{t_l}{2} - p \\
&= \frac{t_l}{2} + 2 \left( \frac{(k-1)\, t_l}{2} - \sum_{p=1}^{k-1} p \right) \\
&= \frac{t_l}{2} + 2 \left( \frac{(k-1)\, t_l}{2} - (k-1)\frac{k}{2} \right) \\
&= \frac{t_l}{2} + (k-1)(t_l - k)
\end{aligned}
$$

**Merging speed-up bound**    We roughly estimate the upper bound of the speed-up we can achieve by merging tokens in a $L$-layer transformer model. Therefore, we only consider attention due to its quadratic scaling with $t_l$. We disregard additional effects reducing speed-up, such as merging overhead, to estimate the upper bound. Further, we assume merging half of the tokens in each layer. The attention in the first layer is unaffected by merging, as we apply token merging between the attention and MLP.

$$
\begin{aligned}
\text{speed up} &\leqslant \frac{L\, t^2}{t^2 + \left(\frac{t}{2}\right)^2 + \left(\frac{t}{4}\right)^2 + \cdots + \left(\frac{t}{2^{L-2}}\right)^2 + \left(\frac{t}{2^{L-1}}\right)^2} \\
&= \frac{L}{\sum_{p=0}^{L-1} \left(\frac{1}{2^p}\right)^2} \\
&= \frac{L}{\sum_{p=0}^{L-1} \left(\frac{1}{4}\right)^p} \qquad \text{using geometric series } \sum_{s=0}^{S} v^s = \frac{1 - v^{S+1}}{1 - v} \text{ for } v \neq 1 \\
&\Rightarrow \frac{L\left(1 - \frac{1}{4}\right)}{1 - \left(\frac{1}{4}\right)^L} \\
&= 3\, L\, 4^{L-1} \cdot (4^L - 1)^{-1}
\end{aligned}
$$

## B.2. Inductive biases

Time series feature domain-specific characteristics, including temporal causality, long sequences, periodicity, trends, and sparsity in the frequency domain. Here, we discuss the interplay of these properties with our token merging algorithm.

Our local merging algorithm is specifically designed to exploit two core properties of time series: First, it preserves temporal causality, as real-world time series are generated by causal processes. Second, it maintains linear complexity as time series often consist of very long token sequences (Godahewa et al., 2021; Grešová et al., 2023). This way, we design a very universal token merging scheme, applicable to many model architectures and datasets, as we show in our experiments.

We conduct new investigations where we trace the tokens being merged throughout the transformer model and show that token merging can exploit periodicity and trends without explicitly modeling these inductive biases. As illustrated in figure 8b, our global merging for time series combines local and global information. However, we did not implement these properties as hard inductive biases to maintain the universality of our algorithm: This way, token merging also performs well on sequential data that does not exhibit trend or periodicity, such as DNA sequences (Grešová et al., 2023), as we show in section 5.4. Stock prices typically also do not have regular periodic patterns. Further, introducing a periodic bias to the neighborhood of our local merging algorithm would break causality, making it inapplicable to decoders.

Autoformer and FEDformer transform the tokens to the frequency space. Autoformer specifically focuses on the autocorrelation. Here, our token merging natively exploits sparsity in the frequency domain and autocorrelation space.

Our token merging algorithm can exploit inductive biases for time series, including periodicity, trends, and sparsity in the frequency or autocorrelation space, but it is not limited to those. This way, it is universally applicable to many architectures and datasets. Further, it features causality and low complexity as inductive biases for 1d-sequence processing.

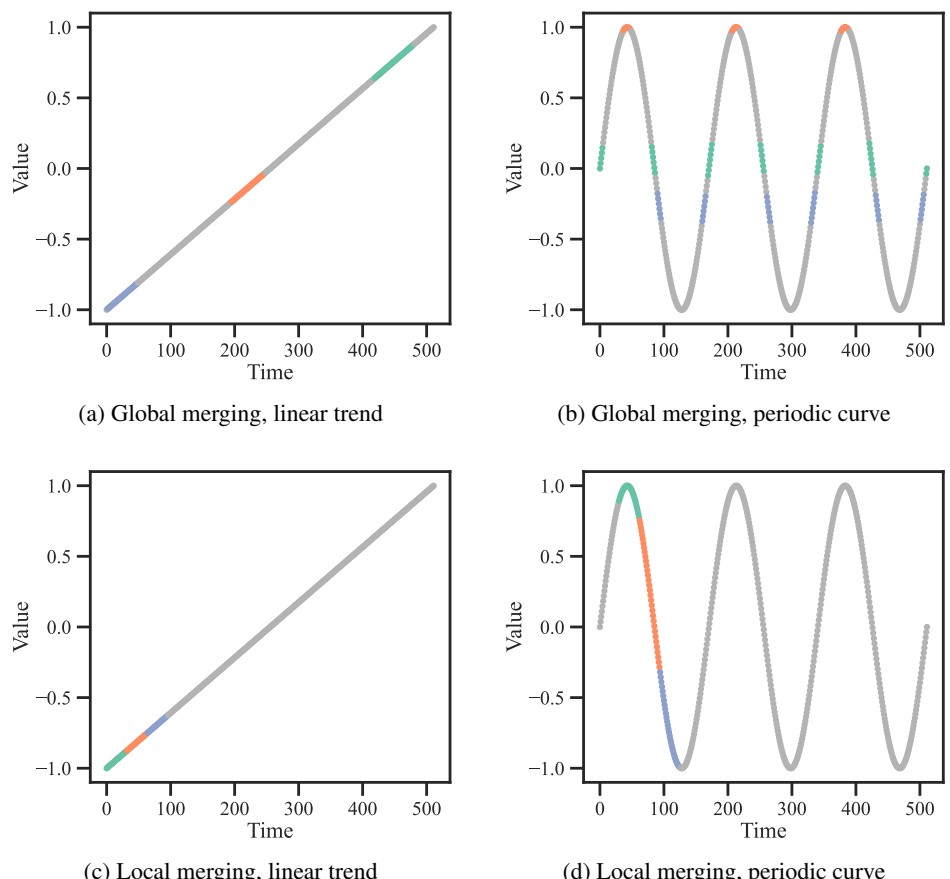

(a) Global merging, linear trend

(b) Global merging, periodic curve

(c) Local merging, linear trend

(d) Local merging, periodic curve

Figure 8: Global and local merging in Chronos base on data with linear trends and periodic patterns. Time series samples merged into the same tokens throughout the transformer are visualized in the same color (top 3 tokens displayed). Local merging preserves locality and causality. Global merging combines local and global information, capitalizing on periodic patterns.

## C. Experiments

Here, we list additional information concerning our experimental settings and resources.

**Datasets** We base our experiments on 5 commonly used multivariate time series datasets covering different forecasting applications: *ETTh1* and *ETTm1* consist of 7 variates measuring the power load and temperature of electric transformers in hourly and quarter-hourly granularity (Zhou et al., 2021). *Weather* consists of 21 meteorological quantities, such as air temperature, and is recorded every 10 minutes in 2020.[1] *Electricity* measures the energy demand of 321 consumers every hour (Godahewa et al., 2021). *Traffic* consists of 862 sensors in the San Francisco Bay Area measuring the road occupancy hourly (Godahewa et al., 2021). We use the same data splits for training, validation, and test as Wu et al. (2021) for consistency.

Since the Chronos foundation model operates univariately and requires considerable computational resources, we randomly sample the same 7000 time series from the test set for all Chronos evaluations. For the ETTh1 dataset, we do not observe relevant differences when comparing the results to the full test set in figure 9.

To explore token merging in an additional sequence-based domain and on a second task, we use the *Dummy Mouse Enhancers Ensembl* dataset (Grešová et al., 2023) for classifying genomic data. It contains very long sequences of nucleotides from a mouse.

**Applying token merging** In our experiments, we generally find it beneficial to allow self-attention to transfer information between tokens before merging them. Therefore, we apply token merging between self-attention and the MLP in all transformer encoders as Bolya et al. (2023). Many transformers exhibit quadratic attention, imposing considerable computational cost. As a result, we do not find the token merging algorithm to introduce a substantial additional overhead. Thus, we choose $k = t_l/2$ to profit from a global merging pool for transformer encoders. For our main experiments, we also apply our causal local merging with $k = 1$ in the transformer decoders between self-attention and cross-attention and finally unmerge all decoder tokens. Therefore, we utilize different merging strategies in transformer encoders and decoders. In architectures utilizing additional tensors like attention masks or positional biases, we merge them using the same correspondences.

In state-space models, we merge tokens after the Hyena or Mamba operator and choose $k = 1$ to not introduce an operation with quadratic complexity into the architecture.

**Hyperparameter optimization** For each transformer architecture, model size, and dataset, we train 32 models without token merging doing hyperparameter tuning of *learning rate* and *dropout* using HEBO (Cowen-Rivers et al., 2022). Here, we apply token merging during inference-time only. We choose the best model based on its validation MSE. We train 17 models with the found hyperparameters, the minimum possible $q_{train}$, and different uniformly spaced $r_{train}$ until all tokens are merged. We again choose the best model based on the MSE for further evaluation. We do 185 hyperparameter optimization trials of both chosen models, trained with and without token merging, using HEBO to find token merging inference hyperparameters $r_{test}$ and $q_{test}$ on the validation set. Please note that $r$ and $q$ might be different for local merging in the encoder and causal local merging in the decoder. Finally, we evaluate once on the test set to report our results.

**Hyperparameters** In table 6 we list the most relevant hyperparameters we used for training the transformer models, including the vanilla Transformer, Autoformer, FEDformer, Informer, and Non-stationary Transformer. For training and testing HyenaDNA (Nguyen et al., 2023) and for testing Chronos (Ansari et al., 2024), we used their default hyperparameters.

**Reproducibility of measurements** We report all results on the same Nvidia A6000 GPU. For training, we utilize Nvidia V100 and A100 GPUs. We measure the end-to-end inference time of the models using 2 warm-ups and 2 measurement runs per batch. The standard deviation of the inference time is generally $< 2\%$ in our experiments. Besides the inference time as practically most relevant quantity, we report FLOPs as a more hardware-independent measure using the thop library (Zhu, 2022). We choose the maximum possible batch size and standardize the results.

**Computational effort** We estimate the computational effort for reproducing our experiments in table 7. Please note that we base some of our experiments on model checkpoints acquired in previous experiments.

---

[1]https://www.bgc-jena.mpg.de/wetter/

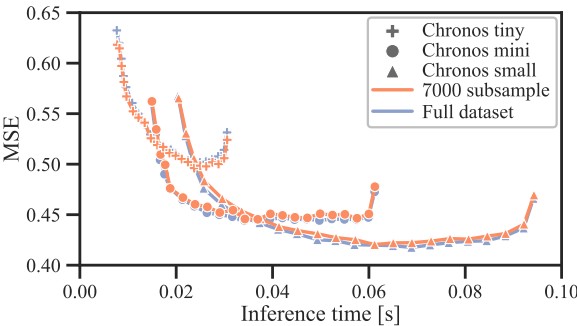

Figure 9: Comparison of Chronos models on the subsampled ETTh1 dataset to the full dataset.

Table 6: Hyperparameters for training the transformer models.

| Hyperparameter | Value |
|---|---:|
| **Training** | |
| Seed | 2024 |
| Optimizer | Adam (Kingma & Ba, 2015) |
| Learning rate | Search space loguniform$[10^{-6}, 10^{-2}]$ |
| Learning rate decay | Exponential, $\gamma = 0.97$ |
| Dropout | Search space uniform$[0.0, 0.25]$ |
| Batch size | 32 |
| Epochs | 100 |
| Early stopping patience | 7 |
| Loss | MSE |
| **Model** | |
| Input length | $m = 192$ |
| Prediction horizon | $p = 96$ |
| Token dimension | $d = 512$ |
| Encoder layers | $L \in \{2, 4, 6, 8, 10\}$ |
| Decoder layers | 1 |
| Attention heads | 8 |
| MLP hidden dimension | 2048 |
| Activation | GELU |

Table 7: Computational effort to reproduce our experiments.

| Experiment | Accelerator | GPU hours |
|---|---|---:|
| Local merging in pretrained models | A6000 | 100 |
| | V100 | 6720 |
| Local merging during training | A6000 | 50 |
| | V100 | 3840 |
| Scaling to large models | A6000 | 500 |
| Token merging in state-space models | A6000 | 40 |
| | A100 | 12 |
| Dynamic token merging | A6000 | 140 |
| Improvement of forecasting quality | A6000 | 30 |
| Dependencies on input length | A6000 | 80 |
| Redundancy of input tokens | A6000 | 5 |

# D. Results

Here, we show additional results for our main experiments.

## D.1. Scaling to large models

In this section, we show complete results on applying token merging to Chronos, a time series foundation model.

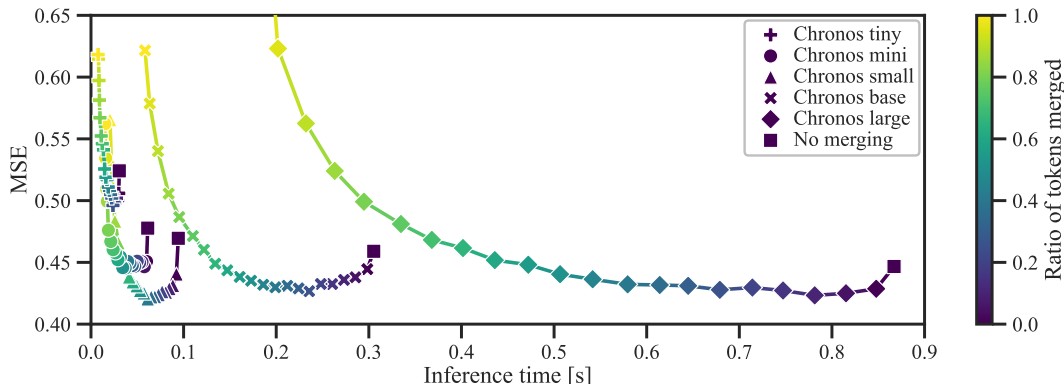

Figure 10: Token merging in different Chronos models on ETTh1.

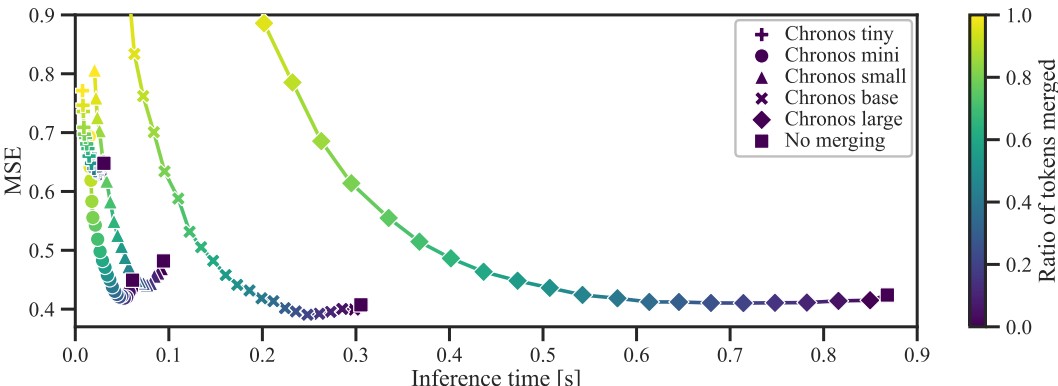

Figure 11: Token merging in different Chronos models on ETTm1.

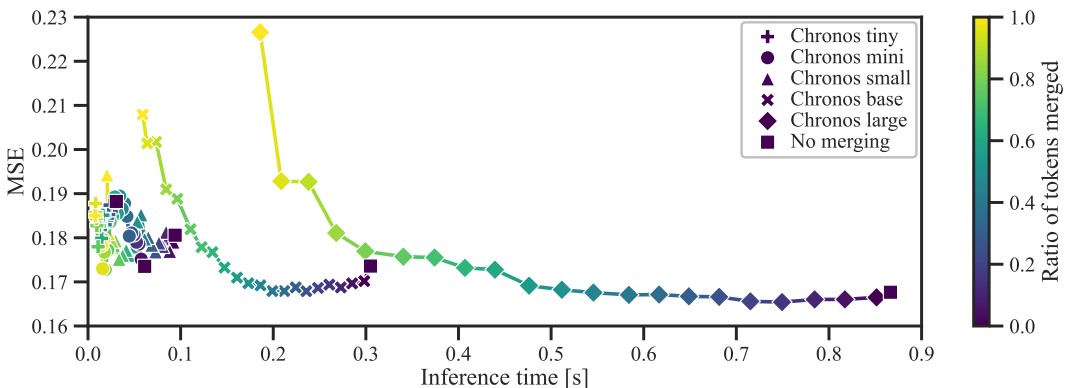

Figure 12: Token merging in different Chronos models on Weather.

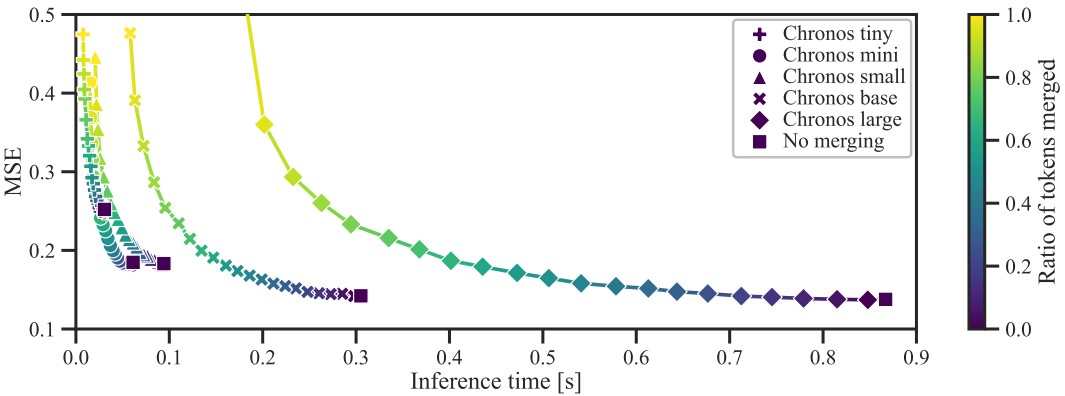

Figure 13: Token merging in different Chronos models on Electricity.

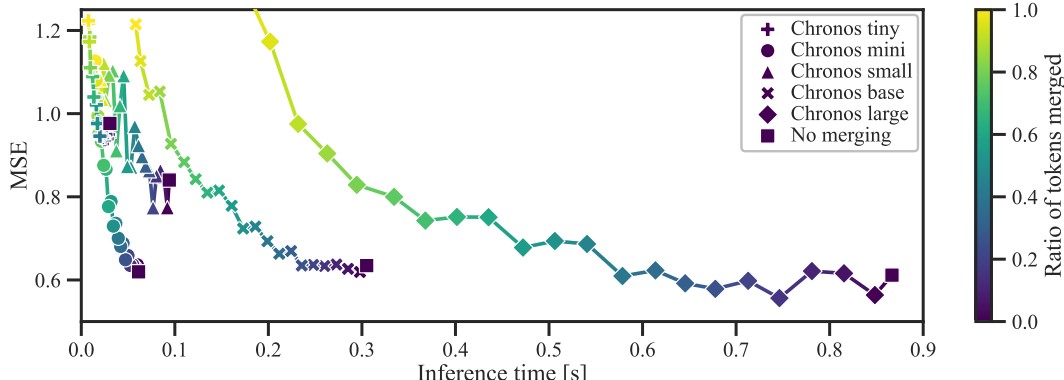

Figure 14: Token merging in different Chronos models on Traffic.

# E. Further investigations

Here, we show additional experiments and results.

### E.1. Token similarity measures

Different distance measures can be utilized to determine similar tokens for merging. Here, we explore the $L_1$ and $L_2$ norms as magnitude-aware metrics and the cosine similarity measuring the angular distance. Our results show that the cosine similarity outperforms both the $L_1$ and $L_2$ norms marginally. Bolya et al. (2023) further ablate the similarity metric for the vision domain.

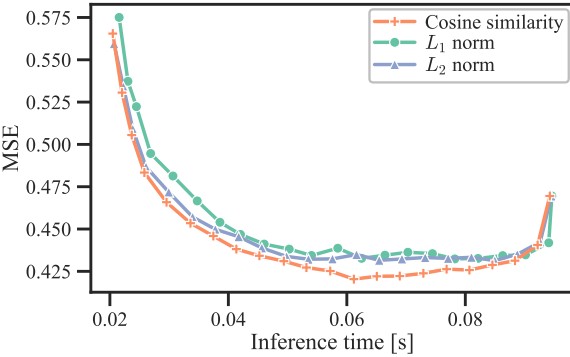

Figure 15: Different token similarity metrics in Chronos small on ETTh1.

### E.2. Token pruning

Here, we compare merging tokens to pruning tokens. Generally, pruning is associated with a higher loss of information. This is also evident in our results in figure 16, where local merging outperforms local pruning.

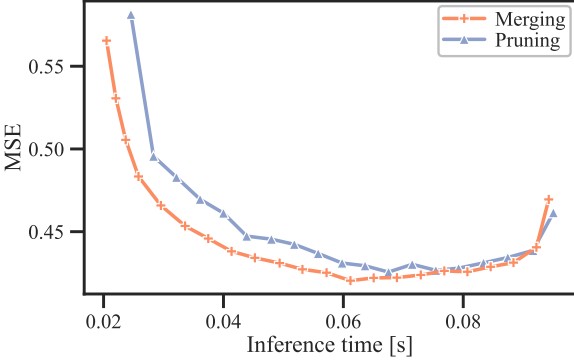

Figure 16: Local merging retains more information compared to local pruning, resulting in better MSE of Chronos small on ETTh1.

### E.3. Tokenization methods

Tokenization involves splitting the input time series into smaller units and embedding them in high-dimensional space. In our experiments, we utilize varying token representations: Transformer architectures in section 5.1 leverage multivariate tokens and a continuous embedding space, while Chronos utilizes univariate tokens embedded into discrete space. Autoformer and FEDformer transform tokens to the frequency domain. In table 8, we further include PatchTST (Nie et al., 2023), which embeds fixed-length subsequences as tokens. While most architectures generate tokens from time series, Hyena and Mamba tokenize nucleotide sequences. We argue that the tokenization method is of minor importance for token merging. Throughout all of our experiments, local merging consistently performs well on top of all token types.

Table 8: Local merging accelerates PatchTST models. Speed-ups are in line with table 1 even though there are only 24 tokens available for merging due to patching.

| Dataset | Layers $L$ | PatchTST | | |
|---|---|---|---|---|
| | | MSE | Accel. | $\text{MSE}_\Delta$ |
| ETTh1 | 2 | 0.37 | 1.17× | 2 % |
| ETTm1 | 2 | 0.30 | 1.17× | 2 % |
| Weather | 2 | 0.16 | 1.98× | 5 % |

### E.4. Improvement of forecasting quality — selective low-pass filter

We find token merging to have a smoothing effect, improving MSE, and show our results on all datasets here.

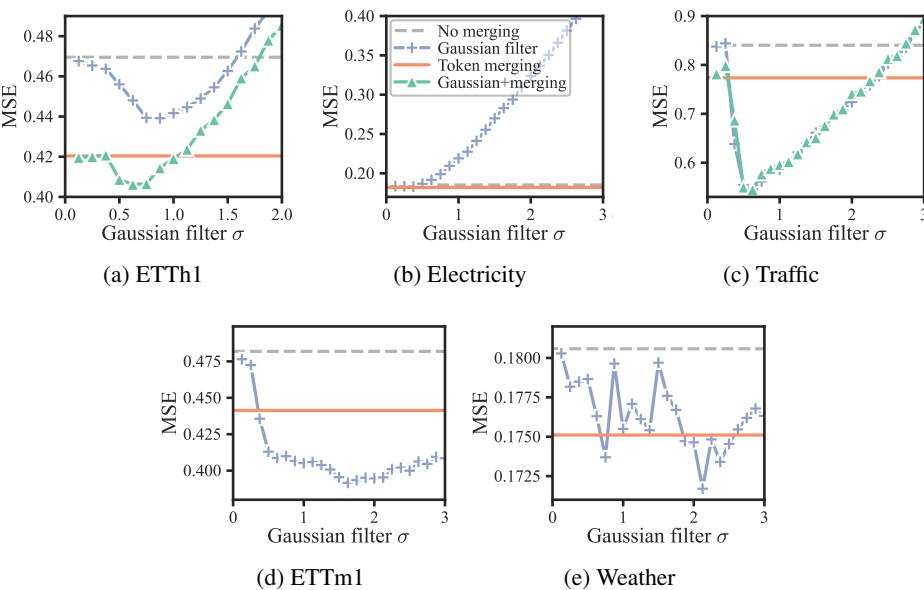

(a) ETTh1    (b) Electricity    (c) Traffic

(d) ETTm1    (e) Weather

Figure 17: Comparing token merging to smoothing the input time series of Chronos small on different datasets.

## E.5. Improvement of forecasting quality — dataset properties

In the following, we show the frequency spectrum of ETTh1, ETTm1, Weather, Electricity, and Traffic datasets.

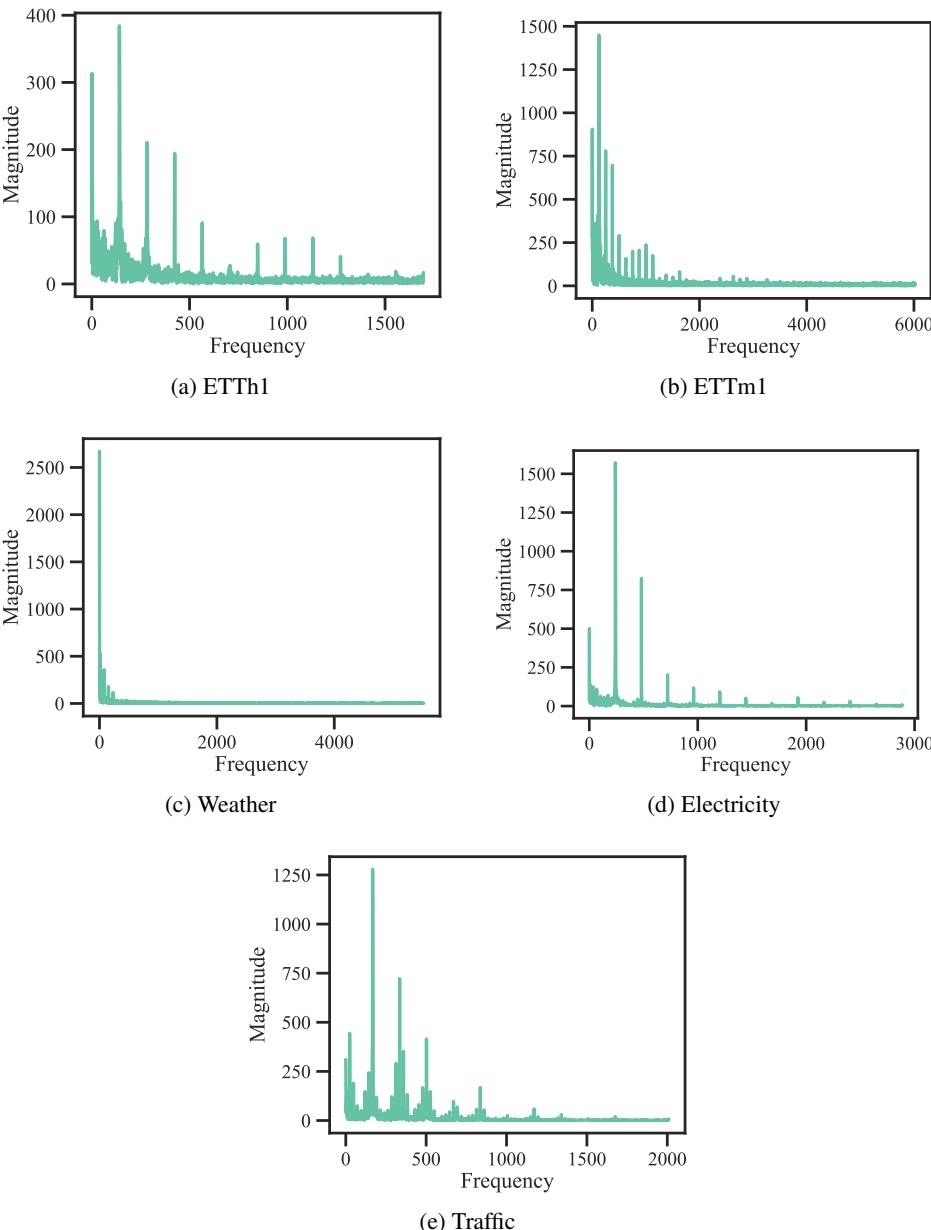

Figure 18: Frequency spectra of different datasets.

### E.6. Redundancy of input tokens

Token merging exploits similarities in data. Intuitively, the number of tokens that can be merged without affecting predictive performance should depend on the redundancy of the tokens. We explore factors influencing the redundancy of input tokens, including their number and positional embeddings. In the following, we use Autoformer's time stamp positional embedding for our ablation.

First, we investigate whether scaling the number of input tokens increases average redundancy on the ETTh1 dataset. As demonstrated in figure 19, the same relative number of tokens is merged for a given merging threshold, independent of input length. Therefore, we suggest scaling the number of merged tokens in each layer $r$ linearly with the input length. Positional embeddings add information about the location of a token within a sequence. As a result, two identical tokens without positional embeddings may show considerable differences when positional embeddings are included, potentially preventing merging. However, figure 19 shows that this effect on token merging is only marginal.

It is worth noting that the attention of the transformer acts as a high-dimensional low-pass filter, effectively generating more redundancy throughout the transformer layers, as Marin et al. (2021) show. Therefore, token merging not only relies on redundancy in the input data but also exploits redundancy that is introduced by the transformer itself.

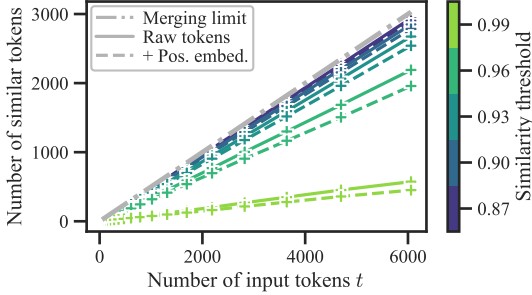

Figure 19: Relative number of redundant tokens for different similarity thresholds on ETTh1 with and without added positional embedding.

### E.7. Dependencies on input length

Here we show an additional evaluation on applying token merging in Chronos models with different input lengths. It is beneficial to choose a larger input length with token merging over a smaller one without.

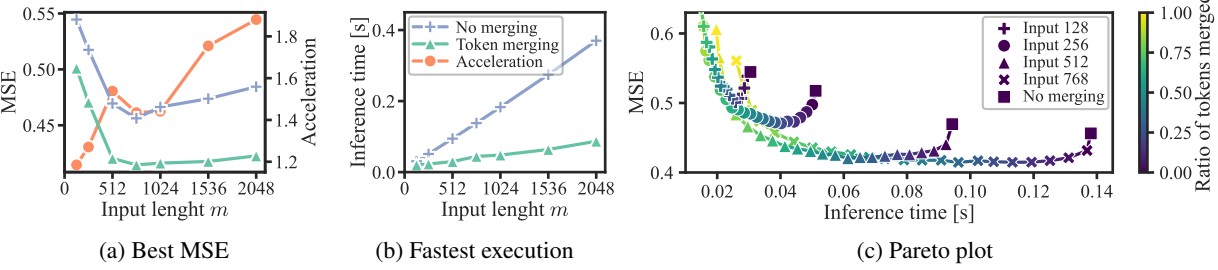

(a) Best MSE      (b) Fastest execution      (c) Pareto plot

Figure 20: Varying the input length of Chronos small on ETTh1.

