# OpenReview forum: "Efficient Time Series Processing for Transformers and State-Space Models through Token Merging"
_ICML.cc/2025/Conference — ICML 2025 poster_

### Official Review · Reviewer_6Amp · 2025-03-05

**Overall Recommendation:** 3

**Summary:**

This paper introduces ​local token merging, a novel algorithm for accelerating time series processing in transformers and state-space models (SSMs). A domain-specific token merging method that computes token similarities within a constrained neighborhood (size k), reducing complexity from quadratic to linear (when k=1). This enables scaling to long sequences while preserving locality and causality. The work bridges token merging—previously limited to vision—to time series, addressing efficiency challenges in long-sequence modeling.

**Claims And Evidence:**

Most claims are well-supported.
Speedup metrics (Table 1, 2, 5) and FLOPs analysis (Appendix B.1) validate complexity reduction. Successful application to Chronos (decoder-only) and encoder-decoder models (Table 2) supports causality claims.

Issues: While Figure 7 shows slight improvements, the paper does not compare dynamic merging to fixed r in batch settings, leaving practical trade-offs unclear. The upper bound in Section B.1 assumes merging half the tokens per layer, but real-world speedups depend on hardware and implementation (e.g., merging overhead).

**Essential References Not Discussed:**

​Adaptive Token Reduction: Liu et al. (2021, DynamicViT) dynamically prunes tokens in vision; comparison could highlight trade-offs between pruning/merging.
​Time Series Tokenization: Nie et al. (2023, PatchTST) uses patching for efficiency; discussing how merging complements patching would contextualize contributions.
​Recent SSMs: Mamba (Gu & Dao, 2023) achieves linear-time inference; analyzing merging in Mamba could strengthen SSM evaluations.

**Experimental Designs Or Analyses:**

Results on 7000 samples may not generalize to full test sets.
The ablation (Figure 9) uses only Autoformer; broader validation across architectures would strengthen conclusions.

**Methods And Evaluation Criteria:**

Local merging is well-motivated for time series, where locality and causality are critical. Restricting merging to neighborhoods (k) balances redundancy exploitation and computational cost.

Chronos experiments use a subsampled test set (7000 series). While practical for compute constraints, full-test-set validation would strengthen claims.

**Other Comments Or Suggestions:**

Page 5, Table 1: “MSE△” formatting inconsistencies.
Section 5.4’s spectral analysis could better differentiate signal entropy and noise

**Other Strengths And Weaknesses:**

Strengths:
First token merging method for time series, enabling causal merging and SSM acceleration.
Massive speedups (54×) in Chronos, relevant for real-world deployment.
Connects spectral properties to merging efficacy, providing actionable insights.
Weaknesses:
No evaluation in batched settings or comparison to threshold-based pruning.
 Lack of code/details for SSM merging.
Limited to Autoformer; broader validation needed.

**Questions For Authors:**

My main concern will lie on the kind of incremental motivation that applies token merging technique from CV to time series processing. What is the main gap between the application of token merge in these two domains (definitely existing), and how has this paper resolved this core issue?  Tehse clarifications should be highlighted.
How does dynamic merging perform in batched inference? Would variable token counts per batch element hinder practical usage?
Could local merging accelerate Mamba? Testing this would broaden SSM applicability.
Would results hold on the full ETTh1 test set?

I am willing to raise my final rating after the rebuttal phase if the authors well-resolved my concerns.

**Relation To Broader Scientific Literature:**

The key contribution of this paper is a technique switch from CV to time series processing.

**Theoretical Claims:**

The derivation of local merging complexity (O(t + (k−1)(t−k))) is correct. The upper-bound speedup estimate for L-layer transformers is reasonable but idealized (ignores layer-wise overhead).

---

> ### Author Rebuttal · Authors · 2025-04-01
>
> Dear Reviewer 6Amp,
>
> Thank you for taking the time to read our paper and for your valuable questions. We are happy to answer them in the following. To this end, we conducted 5 new experiments. Please find anonymous results here: https://figshare.com/s/679d2c1d825228385b2d
>
> **Q:** Gap between token merging in CV and time series. \
> **A:** Prior work in CV only used quadratic token merging in non-causal transformer encoders. We design token merging for time series:
> - Long sequences: Time series often consist of very long sequences, as opposed to CV (L122). Here, quadratic global merging introduces a significant computational burden. We propose local merging with linear complexity. (tab. 5, 16k long sequences).
> - Decoders: In CV, token merging is applied to non-causal encoders. However, for time series, endoder-decoder or decoder-only models are common. Local merging is the first causal merging scheme for decoders. Further, causality is a good inductive bias for time series (tab. 5, L417-426), improving MSE. In contrast to images, real-world time series are generated by causal processes.
> - Forecasting: In CV, most works focus on classification where only 1 cls token needs to be preserved. We use forecasting as a dense task where reducing tokens might be more difficult. To this end, we propose causal unmerging to restore a required number of output tokens.
> - SSM: Prior work focuses on transformers. We are the first to merge tokens in ssm.
> - We propose dynamic merging to mitigate the issue of dissimilar tokens being merged. Prior work utilized fixed merging rates per layer.
> - Besides these technical aspects, our detailed analysis of local merging, such as why it improves MSE, is way beyond literature.
>
> We will highlight these differences in our paper
>
> **Q:** Dynamic merging in batch settings. \
> **A:** Thank you for this nice suggestion. We conducted new experiments for dynamic merging in batched settings. To retain rectangular tensors, we average the number of tokens to be merged among the elements in a batch. For batch size 10 dynamic merging is still better than fixed r merging. Variable token counts per batch element do not hinder practical usage.
>
> **Q:** Would results hold on the full ETTh1 test set? \
> **A:** We subsample 7000 time series for chronos experiments. We have made additional efforts to compare results of the subsampled test set to the full ETTh1 test set. Results also hold for the full set.
>
> **Q:** Section B.1 upper bound for speed up ignores overhead. \
> **A:** You are correct. As the overhead varies between models and implementation, we derive this upper bound theoretically. For our experiments, however, we report the execution time rather than the FLOPs as it includes all overheads.
>
> **Q:** Fig. 9 uses only Autoformer. \
> **A:** This is a good point. For this ablation, we only borrow the trained position embedding from Autoformer. It is quite similar to the one Transformer, Informer, FEDformer and Non-stationary use. The actual model has no effect. We therefore hope to make broad statements.
>
> **Q:** Merging and pruning. \
> **A:** This is a great addition. We conducted new experiments to compare merging and pruning, which is used by DynamicViT. Merging results in better MSE, as it retains more information, and comparable speed up. Further, as opposed to DynamicViT, our pruning does not require any training and is zero-shot-applicable.
>
> **Q:** Tokenization and token merging \
> **A:** In our experiments, the models use 3 different types of tokenization:
> - Models in tab. 1 use multivariate tokens
> - Chronos uses a discrete vocabulary
> - Hyena builds tokens from nucleotides
>
> Autoformer and FEDformer further use the frequency domain. Local merging works on top of all token types. We argue that the tokenization method is of minor importance. We conduct new experiments to include PatchTST. Preliminary results show, that local merging also works on patches. \
> **Technical differences:** Patching compresses only the input data and requires model training (architectural change). Token merging is a general method that accelerates a variety of models in zero-shot setting. It exploits redundancy in every layer (L807-809, transformers generate redundancy).
> Thanks for pointing us to this missing detail.
>
> **Q:** Mamba as SSM \
> **A:** Thank you for this suggestion. We added Mamba for this rebuttal. Local merging achieves over 4x acceleration with only 1.6% drop in accuracy.
>
> **Q:** Lack of details for SSM merging \
> **A:** We are happy to include more details. Do you have specific aspects in mind? We already analyzed the wall-clock time in more detail: The similarity computation of local merging adds 14% of additional wall-clock time to each hyena block, while global merging takes additional 68%. This highlights the value of linear complexity.
>
> Thank you also for pointing us to sec. 5.4 spectral analysis. We will clarify "signal entropy" and "noise".
> We are happy to further discuss any open questions.

---

> > ### Comment · Reviewer_6Amp · 2025-04-02
> >
> > Thank you for addressing the feedback and incorporating additional simulations, which provide valuable empirical support to the methodology. Nevertheless, regarding the pivotal technical concern about adapting token merging mechanisms from computer vision (CV) to time-series analysis, the authors' responses still lack substantial technical depth and fail to clarify fundamental theoretical discrepancies.
> >
> > Specifically, the critical distinction between 2-D patch-level image token merging—which inherently balances local-global semantics through spatial hierarchies—and 1-D vector-based token merging in time series remains underexplored. While CV methods exploit geometric correlations across image patches, temporal sequences exhibit distinct properties like periodicity, trend continuity, and stochastic volatility. This raises questions about whether the proposed adaptation truly addresses the unique challenges of time-series tokenization or merely replicates CV paradigms without domain-specific innovation.
> >
> > Moreover, the technical rationale for directly transplanting 2-D patch merging strategies to 1D time-series slices requires rigorous justification. The original mechanism relies on spatial redundancy reduction in images, whereas time-series redundancy often manifests through temporal autocorrelation or frequency-domain sparsity. Without a systematic analysis of how merging operations interact with these temporal characteristics (e.g., phase distortion in aggregated tokens, information loss in non-stationary segments), the claimed efficiency gains risk being contextually decoupled from the core challenges of time-series modeling.
> >
> > In summary, while the empirical results are compelling, the fundamental rationale for this adaptation—spanning theoretical alignment between CV and time-series token merging, as well as operational validity in 1D contexts—remains inadequately justified and demands deeper technical scrutiny

---

> > > ### Author Response · Authors · 2025-04-03
> > >
> > > Dear Reviewer 6Amp,
> > >
> > > Thank you for appreciating our new results and for elaborating on your questions.
> > > We agree, that time series have distinct properties, such as periodicity, trend, and sparsity in the frequency domain.
> > > In the following, we would like to address your questions:
> > >
> > > **Q:** Analysis on the effect of token merging regarding time series specific properties \
> > > **A:** Thank you for pointing us to these details. We conduct an extensive analysis where we find metrics of time series that benefit local merging in sec. 5.4. We think connecting these metrics to the distinct time series properties you mentioned improves our paper:
> > > - The spectral entropy measures how concentrated power is in the frequency domain. Low values indicate strong periodicity, while high values indicate randomness. Further, the spectral entropy directly relates to sparsity in the frequency domain.
> > > - The total harmonic distortion measures the distortion in a signal caused by harmonics. The more a periodic signal deviates from a pure sine wave, the higher the harmonic content and thus the higher the THD.
> > > - To explore stationarity we utilize the Augmented Dickey-Fuller test and report the percentage of stationary variates on commonly used datasets. As almost all variates are stationary, we can not draw meaningful conclusions  regarding the effect of token merging.
> > >
> > > | Dataset | % Stationary variates |
> > > | -------- | -------- |
> > > | ETTm1     | 100     |
> > > | ETTh1     | 100     |
> > > | Traffic     | 99.8     |
> > > | Electricity     | 91.2     |
> > > | Weather     | 100     |
> > >
> > >
> > > Linking the periodicity, sparsity in the frequency domain, and shape of the time series to the spectral entropy and total harmonic distortion makes our analysis more intuitive.
> > >
> > >
> > > **Q:** Time series specific properties we integrate in our token merging method \
> > > **A:** Thank you for pointing us to this. We will include the following discussion on which time series specific inductive biases we introduce in our method in our paper.
> > >
> > > Our token merging mechanism exploits two core properties of time series, which are not present in CV:
> > > - It preserves temporal causality, as real-world time series are generated by causal processes
> > > - It maintains linear complexity as time series often consist of way more tokens than images in CV (Godahewaetal.,2021; Grešováetal.,2023)
> > >
> > > This way, we design a very universal token merging scheme, applicable to many model architectures and datasets, as we show in our experiments.
> > > We conduct new investigations where we trace the tokens being merged throughout the transformer model. This experiment demonstrates that our merging can exploit distinct properties like periodicity and trend. https://figshare.com/s/679d2c1d825228385b2d As shown in the sine-curve example, our global merging for time series also trades off local and global information. However, we did not implement these properties as hard inductive biases to maintain the universality of our algorithm:
> > > - Token merging can exploit trend and periodicity, as our new experiments show, but it is not tied to these properties. This way, it also performs well on sequential data that does not exhibit trend nor periodicity, such as DNA sequences (Grešováetal., 2023), as we show in sec. 5.8. Stock prices typically also  don't have regular periodic patterns.
> > > - Adding a periodic bias to the neighborhood of our local merging algorithm would further break causality. This way it is not applicable to decoders.
> > > - Autoformer and FEDformer transform the tokens to the frequency space. Autoformer specifically focuses on the autocorrelation. Token merging exploits sparsity in frequency domain and autocorrelation space in these architectures, which you correctly pointed out as two other properties of time series. Our universal algorithm can capitalize on this frequency-based sparsity. However, adding a token-level trend or periodicity bias would be suboptimal here, as tokens are already transformed to the frequency domain.
> > >
> > > We therefore see the universality of our algorithm as its strength. It can exploit inductive biases for time series (periodicity, trend, sparsity in frequency or autocorrelation space), but it is not fixed to those. This way, it is applicable to many architectures and datasets. Futher, it features causality and low complexity for long sequences as inductive biases for 1d-sequence processing. We agree that future work can explore more specialized merging schemes tailored to a specific type of time series, such as to periodic series. However, we think there needs to be initial work on universal, broadly applicable token merging for sequences first.
> > >
> > > Thank you for discussing time series specific properties with us. We think this makes our paper more valuable.
> > > We hope we could answer your questions.

---

### Official Review · Reviewer_mSjb · 2025-03-12

**Overall Recommendation:** 4

**Summary:**

This paper introduces a novel local token merging algorithm for time series models aimed at reducing the computational burden of processing long sequences in transformers and state-space models. By merging tokens within local neighborhoods, the method scales the complexity from quadratic to linear while preserving causality—making it suitable even for transformer decoders.

**Claims And Evidence:**

I think that the claims made by the authors are supported by a variety of experiments. The idea of token merging seems to be effective at improving computational efficiency while retaining good performance.

**Essential References Not Discussed:**

I think that while the authors claim that this is the local merging technique, they do not discuss a recent work that integrates both global and local information of the time series using path signatures [1]. I think that this should be included.

[1] Moreno-Pino, Fernando, et al. "Rough Transformers: Lightweight and Continuous Time Series Modelling through Signature Patching." Advances in Neural Information Processing Systems 37 (2024): 106264-106294.

**Experimental Designs Or Analyses:**

Yes, the experiments are sound.

**Methods And Evaluation Criteria:**

The authors benchmarked on standard datasets used in the literature, and the evaluation is thorough.

**Other Comments Or Suggestions:**

None.

**Other Strengths And Weaknesses:**

-

**Questions For Authors:**

Do the authors feel that similar techniques could be used in language, or is token merging something exclusively usable in the context of time series?

**Relation To Broader Scientific Literature:**

I think one of the key contributions of this paper lies in the fact that this can be integrated into pre-trained models like Chronos. I think this type of technique could become a widely used heuristic to be integrated into time series models.

**Theoretical Claims:**

This is mostly an empirical paper, so the authors did not make many theoretical claims.

---

> ### Author Rebuttal · Authors · 2025-04-01
>
> Dear Reviewer mSjb,
>
> Thank you for your valuable feedback and effort.
>
> The "Rough Transformers" paper is very interesting and we will further discuss it in our related work section. Thank you for pointing us to that. We would like to emphasize that we see local merging as a method to accelerate a variety of models without needing to retrain or finetune them. Local merging is not only applicable to transformers but also to state space models. "Rough Transformers" on the other hand is a specifically designed model architecture.
>
> **Q:** Do the authors feel that similar techniques could be used in language? \
> **A:** This is a great point. We do think that local merging might help to boost the efficiency of language models. Our causal merging and causal unmerging is the first token merging technique applicable to decoders, making it very valuable for decoder or encoder-decoder language models. Previous work in vision breaks causality and is not applicable to decoder models as used in NLP. For long contexts, local mergings linear complexity is particularly helpful.
>
>
> Further, we have made several improvements to our work and present new results in our comments to review 6Amp.
> We are happy to further discuss any open questions.

---

> > ### Comment · Reviewer_mSjb · 2025-04-04
> >
> > I thank the authors for their response. I will raise my score, as I believe this can be a valuable technique for the time-series forecasting community.

---

### Official Review · Reviewer_eLzT · 2025-03-15

**Overall Recommendation:** 3

**Summary:**

This paper proposes to apply Token Merging (ToMe), which was originally developed for vision transformers, to time series models.The main difference between the author’s work and standard token merging is the use of local neighborhoods, ie local merging. The original ToMe formulation allowed tokens from different regions to be merged together if they were similar enough, but this paper restricts the similarity comparison to small locl windows or even only adjacent tokens, which they find to be more efficient and better suited to time series modeling. The authors then show that this has similar results to base ToMe on standard time-series datasets and conduct ablations to demonstrate the value of the local merging. Due to the large number of input tokens in time series models, local merging leads to large speedups, with particularly large speedups compared to those shown in the original ToME paper.

**Claims And Evidence:**

Claim 1. Local merging is better suited to time series data, since  is not quadratic complexity and does not rely on computing pairwise similarities between a large number of tokens. The original ToMe method is quadratic complexity.

Evidence: It’s certainly true that the matching step of ToMe has quadratic complexity. What is less clear is whether this actually matters for wall-clock time. I am not particularly convinced it does, since in the original paper, measurements showed that the computation time of this step was negligible, especially compared to self attention. There is no clear analysis that demonstrates this point in the paper besides Table 5, which does not provide much detail.

Claim 2. Local merging allows significant acceleration of time series models without reducing their performance.

Evidence: This is well supported by experiments.

Claim 3. Local merging preserves causality, which is important for time series data.

Evidence: If I understand correctly, “causality” here means that any token T_i is only predictable from tokens directly before it. In this sense, i see that local merging does preserve causality. What is not clear is whether this actually matters for time series data; there is no ablation in the paper that clearly demonstrates this.

Claim 4: Local merging improves forecasting quality by selectively reducing noise.

Evidence: I don’t really see how applying a Gaussian filter demonstrates that Token Merging is reducing noise. I think the experiment here needs a bit more clarity; I elaborate on this in the experiments section.

**Essential References Not Discussed:**

To my knowledge, the main references were all included in the related works section. Some papers that could also be discussed are those that learnably prune tokens but this is not essential.

**Experimental Designs Or Analyses:**

An issue I found is that many of the important plots that support claims in the analysis section were punted to the supplementary material. I think the paper could be better organized so that the core results were included in the main text and the supplement only contained additional results that would be helpful for building intuition.

Main Experiment

The basic setup here makes sense, However, I don’t understand why the “global merging” (or standard ToMe) was not included here, since comparing local merging to it supports the main novelty in the paper. Secondly, I’m confused what “acceleration” means, and what measure is used here - is it wall clock time? Throughput? GFLOPS? All those base numbers should also be reported, and if that would take too much space, the results should be split into multiple tables. Finally, the Token Merging method (and local merging) depend on a parameter r that controls the number of tokens merged per layer. I dont’ see any discussion of this anywhere except in Figure 3; what was the
Ablation of local merging

There isn’t really an ablation study, since there isn’t many components. However, an important experiment to include would be the effect of increasing the neighborhood size k. There is no such analysis in the main text or the supplement which is an issue - the main point of the paper is that locality is an important property and this should be the main focus of the experiments apart from the main result.

Additional understanding of token merging improvement through signal processing.

I think this set of experiments was interesting. The premise of this experiment is that local merging improves forecasting quality, which was not observed in the vision setting but seems to occur according to Figure 2 (which should be much larger, as it is one of the more interesting results of the paper). However, the experimental design to validate this doesn’t make that much sense to me. Applying Gaussian filtering does not demonstrate that this is what token merging actually does, and neither does showing that combining it with Gaussian filtering improves the MSE either. I may be missing something here from the explanation, so please feel free to correct me if my understanding is wrong.

**Methods And Evaluation Criteria:**

The basic setup makes sense, which is testing several state of the art time series models that rely on attention mechanisms and measuring the downstream performance metric (MSE) and whether there is a speedup.

However, it’s not clear what the “acceleration” measure is - is this wall-clock time, throughput, GFLOPS? Given the results from the token merging paper, it’s certainly surprising to see such larger improvements, so some clarity would be good here.

**Other Comments Or Suggestions:**

Typo: L298 "filer" should be filter.

**Other Strengths And Weaknesses:**

Strengths:

The results of the paper are quite impressive. Most works on token merging do not show improvements to the base model, but just a speedup. This paper shows impressive speed-ups and a clear improvement in the downstream metrics. I also think that the idea to analyze the reason why local merging works better is great and if executed correctly would provide some key intuitions for others who use this method.

Weaknesses:

Ultimately this paper is an incremental improvement to the original token merging paper, in the context of applying a known technique to a different domain with a small tweak to the original algorithm; there’s not a lot of novelty besides adding the locality.

While I see the value in local merging for time series data, the change is extremely minimal - the method consists of restricting the window to small local neighborhoods and is a couple-line code change.

This would be a more valuable contribution if it were better motivated. In the original ToMe work, the matching process has almost negligible runtime; computing cosine similarity is extremely cheap. Is the complexity of this operation actually meaningful? Theoretical complexity does not really translate to wall-clock time which is what matters for running these models at scale; there is no analysis that justifies this part.

Finally, as discussed in the experiments section, the experiments and analysis need to be cleaned up and clarified; I think a lot of important details are currently missing. In the rebuttal phase, I would like to see convincing explanations from the authors why the experiments as currently shown are sufficient for justifying the claims made in the paper.

**Questions For Authors:**

Questions:
In Table 1, why does Autoformer and Informer have lots of rows with no speedup? Was token merging not appilcable in these cases? Why or why not?

The gaussian + merging line seems to be missing from Figure 4 b (electricity). Is this intentional?

**Relation To Broader Scientific Literature:**

Token merging showed that for inputs with many redundant tokens, similar tokens could be combined together to get essentially the same results while significantly speeding up the model. This has been shown mostly in computer vision, but also in language models and vision-language case. This paper demonstrates that the effect occurs too in time series models. Furthermore, the paper’s analysis section attempts to explain why local merging actually improves the performance rather than keeping it the same as occurs in other merging works. This paper, if all experiments are properly conducted and the claims completely validate, would provide a clean way to accelerate time series models while also supporting the method with empirical intuitions.

**Theoretical Claims:**

The only theoretical claims are related to the complexity of the proposed methods. These proofs are relatively straightforward and are included in the supplementary material, and seem reasonable to me.

---

> ### Author Rebuttal · Authors · 2025-04-01
>
> Dear Reviewer eLzT,
>
> Thank you for taking the time to read our paper and for your valuable comments. We are happy to answer them in the following.
>
> **Q:** Local merging is a minimal contribution. \
> **A:** We see our main contribution in investigating token merging for time series in great detail. Additionally, we propose new mechanisms to extend token merging to causal decoders, state-space models and long sequences.
> (Please see our Rebuttal on Review 6Amp for a full list of novelties)
>
> **Q:** Does linear complexity translate to wall-clock time? \
> **A:** We think this is a great point. We compare global merging with local merging in sec. 5.8 and tab. 5. Using subquadratic ssm and a very long context of 16k tokens, we showcase the advantage of linear complexity. Computing the cosine similarity of all tokens in global merging has a significant overhead, resulting in only 2.92x speed up while local merging achieves greater acceleration of 3.62x. For this rebuttal, we investigate the wall-clock time in more detail. The similarity computation of local merging adds 14% of additional wall-clock time to each hyena block. For global merging, however, this is an additional 68% of wall-clock time. Thank you for pointing us to this missing detail, we will clarify that in our paper.
>
> **Q:** Is preserving causality important for time series? \
> **A:** Thanks for bringing this up, you understood it correctly, and we will explain it more explicitly. Preserving causality has two important aspects: First, it enables token merging in decoder models, which are commonly used in time series processing. Previous work only focused on encoders. Second,  causality and order seems to be a good inductive bias for time series, as we show in tab. 5. Here, it is better to rely on a smaller merging pool while preserving causality (74% vs. 69% accuracy, L417-418, L424-432). Intuitively, real-world time series are generated by causal processes.
>
> **Q:** Gaussian filtering does not demonstrate that this is what token merging does. \
> **A:** Gaussian filtering smoothes the time series acting, as a low pass filter. We argue that averaging two tokens also reduces noise, but in an adaptive manner as only similar tokens are merged. To validate our intuition, we compare local merging to gaussian filtering on 5 datasets. On datasets where gaussian filtering improves forecasting quality, local merging also improves quality. On Electricity, both gaussian filtering and local merging are not effective (fig. 4, 15). We further validate the low-pass-filtering hypothesis by correlating local mergings MSE improvements to dataset properties related to noise (see Dataset properties in sec. 5.4). Please note that we try to show that local merging acts as an adaptive filtering algorithm. Not that it is the same as gaussian filtering. We agree that our comparison is not absolute, but we gather lots of evidence that local merging can be compared to gaussian filtering.
>
> **Q:** Table 1, Autoformer and Informer have rows with no speed up. \
> **A:** In these experiments, we apply token merging only during inference. This disturbs some models too much. Thus, we report the results without merging (see L228-233, L251-255). However, in sec. 5.2 we show that we can accelerate these models when applying token merging during training. This way the models can adapt to token merging.
>
> **Q:** The gaussian + merging line  missing from Figure 4 b. \
> **A:** This is intentional, as you correctly guessed, the gaussian + merging line was exactly following the gaussian line.
>
> **Q:** Acceleration measure?\
> **A:** Thank you for pointing this out. We base our acceleration on the wall-clock time (see L703) as we think it is the practically most relevant measure. However, not reported in the paper, we notice even larger accelerations in FLOPs in all experiments as they do not measure GPU overhead.
>
> **Q:** Important plots in the appendix. \
> **A:** Due to the page limit we moved many results to the appendix. We will  restructure the final paper.
>
> **Q:** Ablate neighborhood size k. \
> **A:** In our preliminary experiments we find that either k=1 (linear complexity, causality, locality bias) or k=tl/2 (global merging pool) leads to best results. In many transformer encoders that already exhibit large computational cost, we utilize k=tl/2 to profit from a global merging pool (L679-683). In decoders, ssm, and for long sequence processing we find local merging with k=1 to work best (sec. 5.8). Other values for k were suboptimal. We will add this crucial information to the paper.
>
> **Q:** How we chose hyperparameter r. \
> **A:** We do 185 hyperparameter optimization trials to find the best r (see L692-694) satisfying our tolerated MSE drop (see L227-230) for every model. In fig. 2,3,4,5,7 r is plotted as the color bar and the different points in the diagram.
>
> Thank you for also finding a typo.
> We are happy to further discuss any open questions.
> We present new results in our comment to review 6Amp.

---

> > ### Comment · Reviewer_eLzT · 2025-04-05
> >
> > Thank you to the authors for taking the time to address my concerns. Unfortunately, I will retain my rating at Weak Reject.
> > The main issue with the paper is that the contribution is minimal. The paper makes a small modification to token merging and applies it to a new domain, without much new insights.
> >
> > In the rebuttal, the authors state that their main contribution is that it applies ToMe to a new domain. However, the experiments of the paper are not really focused on this - they are more focused on understanding and measuring local merging.
> >
> > The authors responded to my wall-clock time concerns by pointing to L703 which states that they focus on FLOPS since it is hardware agnostic. FLOPs are known to be a subioptimal measure of speed - many "efficient" attention methods significantly decrease GFLOPs but lead to no actual wall-clock speedup. The only experimental measure of wall-clock time is (as the authors state) the SSM-focused analysis in Table 5. I think if the paper is focused on efficiency, there should be much more experiments devoted to understanding the speedup and its effect on quality - this is missing from the paper as it is.
> >
> > The authors addressed my concerns about the hyperparameter r, the analysis is located in the supplement.
> >
> > Finally, I think the paper needs serious reorganization. The experiment section is extremely heavy on text and focuses on experiments that do not support or add to the papers focus on time-series analysis. Furthermore, a lot of important details are rlegated to the supplement, resulting in a supplement that contains more useful information than the main paper. The figures and tables in the main text should be much larger and clearer so that readers can easily tell what the conclusions of experiments are and how they support the main claim of the paper. I would not go so far as Reviewer 1 to say that it is sloppy, but I think it could be seriously improved.
> >
> > If the authors have time to respond to this rebuttal comment, I'd be happy to answer any further questions and reconsider my rating as needed.

---

> > > ### Author Response · Authors · 2025-04-07
> > >
> > > Dear Reviewer eLzT,
> > >
> > > Thank you for your questions. We will answer them in the following:
> > >
> > > **Q:** Contribution \
> > > **A:** We summarize novel aspects of our work bellow:
> > >
> > > **Methodological contributions:**
> > > - **Adaption to long sequences:** Time series often consist of very long sequences, as opposed to CV (L122). Here, quadratic global merging introduces a significant computational burden. We propose local merging with linear complexity.
> > > - **Decoder compatible merging:** Existing merging schemes can not be directly applied to causal decoders. We address this limitation and propose local merging as the first causal merging scheme. In our experiments, causality is a good inductive bias for time series (tab. 5, L417-426), improving MSE.
> > > - **Forecasting:** We extend token mergings application from classification tasks in CV to dense forecasting tasks. To this end, we propose causal unmerging to restore a required number of output tokens.
> > > - **SSM:** Prior work focuses on transformers. We are the first to show the effectiveness of token merging in SSMs.
> > > - **Dynamic merging:** We propose dynamic merging to mitigate the issue of dissimilar tokens being merged. This adjusts for possible differences between multiple samples in a batch. Prior work utilized fixed merging rates.
> > >
> > > **New time series specific insights:**
> > > Beyond methodological contributions, we investigated dataset and model specific properties that predict possible benefits from token merging:
> > > - Datasets with high spectral entropy and total harmonic distortion are particularly amenable to token merging.
> > > - Some model architectures learn more similar tokens than others, benefiting local merging.
> > > - In our updated manuscript we now connect these signal processing metrics to periodicity, sparsity in the frequency domain, and shape of the time series (please see our last reply on Reviewer 6Amp)
> > > - We gain more insights in local merging and argue that it has low-pass-filtering effects.
> > >
> > > **Actions:**
> > > Thank you pointing out that our contributions should be more prominently listed. We make the following adjustments:
> > > - List our contributions in the introduction
> > > - In our method, we will have a distinct paragraph for the technical contributions (local merging, causality, unmerging, dynamic merging)
> > >
> > >
> > > **Q:** Wall-clock time \
> > > **A:** We are sorry for this misunderstanding. You correctly pointed out that the wall-clock time is the practically most relevant quantity. This is why we *always* report wall-clock time based accelerations (see L703 "Besides the inference time as practically most relevant quantity, ..."). Dynamic merging in sec. 5.8 is the *only* case where we report FLOPs. Wall-clock time is used for *all* other experiments.
> > >
> > > For the SSM experiment we provided in the rebuttal, we specifically measured the wall-clock time the similarity computation of local vs. global merging takes in a Hyena block. (highlighting this might have caused confusion)
> > >
> > > **Actions:**
> > > We will make it more clear in L703 that almost all accelerations are based on wall-clock time (except dynamic merging).
> > >
> > > **Q:** Restructuring of the paper \
> > > **A:** Thank you for making detailed suggestions on how to improve the write-up of our paper. We are already working on that. Due to the page limit and the large amount of experiments we did we had to resize many figures and move some experiments to the appendix. However, for the final paper, there will be one extra page available for the main text.
> > >
> > > **Actions:**
> > > - We will resize the most important figures and move important details back from the appendix to the main text using the one extra page available
> > > - To restructure our experiments, we will describe each experiment at the beginning of 5. and how it is interconnected to the claims. In every experiment subsection, we will further add a final sentence mapping the result to the respective claim / overarching goal.
> > > - **Our new structure:** We first investigate the effectiveness of local merging in: 5.1) pretrained transformer models, 5.2) during model training, 5.3) in foundation models, and 5.4.) in state space models. \
> > > Next, in 5.5) we investigate how and in which cases token merging has the biggest benefits and find model and dataset specific properties that predict token merging effectiveness. \
> > > Lastly, we explore design choices within our algorithm, such as reducing the sequence length in 5.6) or applying merging dynamically instead of at a fixed rate in 5.7).
> > >
> > > We think that clarifying these aspects will make our paper more valuable. We now present our results in this structured way. We hope that we could address your concerns and think that local merging is a valuable addition for the time series community.

---

### Official Review · Reviewer_HVPc · 2025-03-16

**Overall Recommendation:** 1

**Summary:**

The paper proposes local token merging to improve transformer efficiency. Building up on Bolya 2023, the proposed method appear to compute similar tokens within a local neighborhood (as opposed to all to all) and merge them. There are other techniques mentioned in the text, but the writeup is not organized enough to point it out explicitly -- either in the introduction or in the methods section. The method and experiment description make is really difficult to identify and summarize the proposed concepts.

There seems to an attempt of evaluating on many datasets and compare with many models. But the unorganized and sloppy writeup makes it difficult to identify a single narrative or theme.

**Claims And Evidence:**

The text of the paper makes it difficult to understand a consistent and clear narrative of the claim.

Token merging would reduce the computation of both transformer and state space models. This is intuitive and follows directly from the model design.

The paper seems to be having difficulty taking a single stance on their claims. Lines 243~246, in both columns, seem to claim the proposed local merging improves both accuracy and efficiency. But Lines 354 ~356, some results in Table 1 imply reducing tokens would decrease accuracy.

I would stick to one narrative: either we say token merging reduces model  processing/time and the accuracy is not affected, i.e.. the accuracy drop is tolerable and sometimes even increases; or we say we fix the MSE at a certain level and show how much efficiency we can improve at this level.

Otherwise, these two issues get tangled up very quickly and obscures the message the manuscript is trying to convey, e.g., Lines243~255 right column, Section 5.5., 5.6.

**Essential References Not Discussed:**

---

**Experimental Designs Or Analyses:**

---

**Methods And Evaluation Criteria:**

**Contribution:** I am not even 50% clear what are contributions of the paper. From Section 3, it appears the contribution is local similarity computation as merging. The section does not clarify other important details, e.g., how often tokens are reduced, by how much, and how the tokens were merged. From Line 175 right column, I am guessing the proposed method adopts the exact setting of Bolya 2023 for these. Is restricting the similarity and merging to local neighborhood the main contribution and sets the proposed method apart from Bolya 2023?

It seems like the manuscript also proposed merging technique for decoders, which I am guessing Bolya 2023 does not do as it is encoder only. Is this a novel idea? If so, why was it not describes in the methods section and deferred to supplemental? Is it not something that distinguishes this method from Bolya 2023? Or is it not important enough or incremental.
If it is indeed an important distinction, the experiment section should clearly show empirical evidences of its benefits under a separate section. Now, the reader has to search and find out the the experiments on chronos pertains to encoder-decoder merging technique.

Also, is Dynamic merging another technique that is novel and being proposed here? Then, why is it mentioned in page 7 under experiments section?

**Exposition** The manuscript  appears to be too poor to convey even the appreciable findings or efforts from the authors. Apart from what already mentioned above, the title does not mention local token merging which is probably the primary claim the submission makes. Figure 1 confuses the readers more than it clarifies: are the numbers ids of tokens? If so, the matrix should be 8x8 if I understood correctly. How the different colors are illustrating anything about the technique? Line 056 seems to give an impression this is an analysis paper not a model design proposal. In Section 3 methods, it is not clear how token merging in computer vision is relevant for model description whereas the token merging for decoders, which sounds like novel, is pushed to supplementary.


**Experimental results:** The experiment section looks like dump of "all experiments we did" rather than empirical evidences of the claims we made as contribution. See the comments in section above.

**Other Comments Or Suggestions:**

I would rethink the whole approach, ponder on what is the approach the paper should suggest to improve how we use transformers or state space models for time series and reorganize accordingly. Right now, the only things the the readers take away from the submission is local token merging reduces computation -- which I dont understand why is not obvious already.

**Other Strengths And Weaknesses:**

---

**Questions For Authors:**

---

**Relation To Broader Scientific Literature:**

---

**Theoretical Claims:**

---

---

> ### Author Rebuttal · Authors · 2025-04-01
>
> Dear Reviewer HVPc,
>
> Thank you for taking the time to read our paper. We would like to address your concerns in the following:
>
> **Q:** Write-up of the paper \
> **A:** We are sorry, that you were confused by the write-up. We will rework the writeup of our Methods and Introduction section to point out our contributions more clearly.
>
> **Q:** Contributions \
> **A:** Previous work in CV only used quadratic token merging in non-causal transformer encoders. We design token merging for time series:
> - Long sequences: Time series often consist of very long sequences, as opposed to CV (L122). Here, quadratic global merging introduces a significant computational overhead. We propose local merging with linear complexity. (see tab. 5, 16k long sequences).
> - Decoders: In CV, token merging is applied to non-causal encoders. However, for time series, endoder-decoder or decoder-only models are commonly used. Local merging is the first causal merging scheme for decoders. Further, causality is a good inductive bias for time series (tab. 5, L417-426), improving MSE. In contrast to images, real-world time series are generated by causal processes.
> - Forecasting: In CV, most works focus on classification where only 1 cls token needs to be preserved. We, however, use forecasting as a dense task where reducing tokens might be more difficult. To this end, we propose causal unmerging to restore a required number of output tokens.
> - SSM: Previous work only focuses on transformers. We are the first to merge tokens in ssm.
> - We propose dynamic merging to mitigate the issue of dissimilar tokens being merged. Previous work utilized fixed merging rates per layer.
> - Besides these technical aspects, our detailed analysis of local merging, such as why it improves MSE, is way beyond literature.
>
> Due to the page limit and the large amount of experiments, we had to move some aspects to the appendix. However, that seems to cause confusion. We will list our contributions more clearly in the Method and Introduction section.
>
> **Q:** No experiments regarding causality \
> **A:** Preserving causality has two advantages. First, it enables token merging in transformer decoders. Our main experiment in tab. 1 therefore demonstrates causal merging implicitly as we merge tokens in the encoder and the decoder. Second, causality is a good inductive bias for time series. We show that when comparing local merging with k=1 to global merging in sec. 5.8 and tab. 5. Global merging can exploit a larger merging pool with more similar tokens. However, local merging is still better due to its causal locality bias (74% vs. 69% accuracy, L417-418, L424-432). Reviewers mSjb and 6Amp find our causality claim well supported. Thank you for pointing us to that. We will mention the link to causality more explicitly in these experiments.
>
> **Q:** The paper seems to be having difficulty taking a single stance on their claims. Lines 243~246, in both columns, seem to claim the proposed local merging improves both accuracy and efficiency. But Lines 354 ~356, some results in Table 1 imply reducing tokens would decrease accuracy. \
> **A:** We investigate token merging from different points of view. Sometimes token merging improves MSE while accelerating the model at the same time. This is mostly true for chronos models in sec. 5.3 and L82-89. In other settings, token merging accelerates the model at some cost of MSE. This is mostly true for the models in tab 1. We find empirical and theoretical explanations for these different behaviors in sec. 5.4 and 5.5. We think that an evaluation from both points of view (finding the best MSE and finding the fastest model at some MSE cost) is very valuable. Both settings relate to different practical applications, i.e., finding the best or the fastest model. We argue that our objective evaluation is therefore especially valuable, rather than focussing on a single setting. We try to disentangle both settings in the write-up of the final paper.
>
> **Q:** Colors in figure 1 \
> **A:** Thank you for pointing us to this figure. The colors illustrate how different merging distances correspond to entries in the similarity matrix and to different values of k, i.e., which merging correspondences are valid for a given k and which reduced similarity set has to be computed. To avoid further confusion, we will reduce the matrix in fig. 1 to 3x3 (tokens 1 to 6).
>
> We hope we could address your concerns and we are happy to answer any follow-up questions.
> Further, we have made several improvements to our work and present new results in our comments to review 6Amp.

---

> > ### Comment · Reviewer_HVPc · 2025-04-04
> >
> > I have to politely point out that rebuttal is as sloppy as the paper itself. Authors, pls try to look from a reader's point of view and not be offended by a reader's honest opinion.
> >
> > - One of my main concerns, i.e., how is the submission different from Bolya 2023, is not precisely answered in the rebuttal.
> >
> > - The reason for this question: *Q: The paper seems to be having difficulty taking a single stance on their claims. Lines 243~246, in both columns, seem to claim the proposed local merging improves both accuracy and efficiency. But Lines 354 ~356, some results in Table 1 imply reducing tokens would decrease accuracy.* was to understand when a researcher in the field would like to use it. The way I am thinking is, when a researcher/practitioner reads a paper, s/he asks how does the work help solving the problem s/he is dealing with. Will it improve accuracy? Will it make the her method more efficient? From this writeup, as well as the rebuttal, it is difficult to answer conclusively what would be the benefit.

---

> > > ### Author Response · Authors · 2025-04-07
> > >
> > > Dear Reviewer HVPc,
> > >
> > > Thank you for detailing your concerns. We will address them in the following:
> > >
> > > **Q:** Gap between token merging in CV (Bolya 2023) and time series (ours). \
> > > **A:** Bolya 2023 proposed token merging for computer vision. Prior work  only used quadratic token merging in non-causal transformer encoders. We, however, design token merging for time series:
> > >
> > > **Technical differences:**
> > > - Long sequences: Time series often consist of very long sequences, as opposed to CV (L122). Here, quadratic global merging introduces a significant computational burden. We propose local merging with linear complexity.
> > > - Decoders: Existing merging schemes can not be directly applied to causal decoders. We address this limitation and propose local merging as the first causal merging scheme. In our experiments, causality is a good inductive bias for time series (tab. 5, L417-426), improving MSE.
> > > - Forecasting: We extend token mergings application from classification tasks in CV to dense forecasting tasks. To this end, we propose causal unmerging to restore a required number of output tokens.
> > > - SSM: Prior work focuses on transformers. We are the first to show the effectiveness of token merging in SSMs.
> > > - We propose dynamic merging to mitigate the issue of dissimilar tokens being merged. Prior work utilized fixed merging rates.
> > >
> > > **New time series specific insights:** \
> > > Beyond methodological contributions, we investigated dataset and model specific properties that predict possible benefits from token merging:
> > > - Datasets with high spectral entropy and total harmonic distortion are particularly amenable to token merging.
> > > - Some model architectures learn more similar tokens than others, benefiting local merging.
> > > - In our updated manuscript we now connect these signal processing metrics to periodicity, sparsity in the frequency domain, and shape of the time series (please see our last reply on Reviewer 6Amp)
> > >
> > > **Actions:**
> > > Thank you for noticing us that our contributions should be more prominently listed in our paper. We make the following adjustments:
> > > - List our contributions in the introduction
> > > - In our method, we will have a distinct paragraph for the technical contributions (local merging, causality, unmerging, dynamic merging)
> > > - In our experiments, we will clarify to which technical aspect the respective experiment contributes. Further, we will describe each experiment at the beginning of 5.
> > > - **Our new structure:** We first investigate the effectiveness of local merging in: 5.1) pretrained transformer models, 5.2) during model training, 5.3) in foundation models, and 5.4.) in state space models.
> > > Next, in 5.5) we investigate how and in which cases token merging has the biggest benefits and find model and dataset specific properties that predict token merging effectiveness.
> > > Lastly, we explore design choices within our algorithm, such as reducing the sequence length in 5.6) or applying merging dynamically instead of at a fixed rate in 5.7).
> > >
> > >
> > > **Q:** Will it improve accuracy? Will it make the method more efficient? \
> > > **A:** We appreciate your practical approach and we will make the practical benefits more clear.
> > > We observe two extremes in token merging:
> > > - For most models, local merging boosts efficiency at some MSE cost (tab.1).
> > > - On Chronos models and Hyena, we additionally observe improvements in MSE. Here, token merging boosts efficiency and simultaneously improves MSE (faster and better). This results in two interesting settings for practitioners:
> > >     -  Maximum acceleration at an upper-bounded MSE cost
> > >     -  Best MSE while still accelerating the model
> > >     -  We distinguish between both cases in tab.2 and tab.5 naming them "fastest" and "best".
> > >
> > > Overall, token merging leads to pareto optimal efficiency utility trade-offs in all of our experiments.
> > > Further, we can predict local mergings benefit from model and dataset specific properties, as described earlier.
> > >
> > >
> > > **Actions:**
> > > Following our answer above, we will make the two different approaches more clear (Will it improve accuracy and efficiency? Will it improve efficiency at some MSE cost?)
> > > - We define the overarching goal of making models more efficient (in  introduction and beginning of experiments)
> > > - In our experiments, we observe everything from faster and considerably worse to faster and even better.
> > > - Motivated by this novel behavior, we perform our investigation of why local merging can improve MSE in sec. 5.4
> > > - We will add overarching comments such as "local merging boosts the efficiency of every architecture", "on Chronos and Hyena local merging boosts efficiency and simultaneously improves MSE"
> > >
> > > We now present our results in the structured way we described above. We think this makes our paper more valuable, especially due to the large amount of experiments.
> > > We hope we could resolve your concerns. Further, we would like to point you to our new results https://figshare.com/s/679d2c1d825228385b2d

---

### Decision · Program_Chairs · 2025-05-01

**Decision:**

Accept (poster)

**Comment:**

The paper received mixed initial reviews, with one strong reject (#HVPc), one weak reject (#eLzT), and two weak accepts (#mSjb and #6Amp). After a detailed and constructive rebuttal from the authors addressing technical concerns, experimental clarity, and domain-specific adaptation of token merging to time series models, reviewers #mSjb, #eLzT, and #6Amp all raised their ratings to weak accept or accept. Reviewer #HVPc maintained a negative stance, primarily due to concerns about clarity and perceived lack of novelty, but ultimately expressed no objection to acceptance.

After carefully considering the paper, the reviews, and the rebuttal, the AC agrees with the majority opinion and supports acceptance. However, some concerns remain unresolved, particularly regarding the clarity of the presentation and the articulation of the paper’s core contributions. The AC encourages the authors to revise the manuscript thoroughly, addressing all reviewer feedback and improving the exposition in preparation for the camera-ready version.